# A monthly Arctic sea ice thickness product from 1995 to 2023 using multiple radar altimetry data

Feng Xiao[1,2,3], Shengkai Zhang[1,2], Jiaxing Li[1,2], Tong Geng[1,2], Tingguo Lu[1,2], Hui Luo[1,2], Fei Li[1,2]

[1]Chinese Antarctic Center of Surveying and Mapping, Wuhan University, Wuhan, 430079, China

[2]Key Laboratory of Polar Environment Monitoring and Public Governance (Wuhan University), Ministry of Education, Wuhan, 430079, China

[3]State Key Laboratory of Geodesy and Earth's Dynamics, Innovation Academy for

Precision Measurement Science and Technology, CAS, Wuhan, 430077, China

Correspondence: Shengkai Zhang (zskai@whu.edu.cn) and Feng Xiao

(shaw89@whu.edu.cn)

**Abstract.** Arctic sea ice plays a crucial role in studies of regional and global climate

change. Satellite observations have shown that the extent of Arctic sea ice has been declining over the last four decades. However, long-term variations in Arctic sea ice thickness (SIT) have received less attention because SIT cannot be measured directly by satellite-based instruments. Here, we present a monthly Arctic SIT product based on multiple radar altimetry observations from ERS-2, Envisat, and CryoSat-2. To ensure

the accuracy of the SIT retrievals, a novel data processing method is proposed, including leads detection, freeboard conversion to thickness, and inter-mission bias correction. Finally, the monthly SIT estimates for the Arctic Ocean from October 1995 to December 2023 are generated. The thickness estimates are posted on a 5 km resolution polar stereographic grid. The variations in Arctic SIT are analyzed in terms

of spatial and temporal distributions. Furthermore, the SIT estimates are compared with observations from upward-looking sonars and airborne laser altimetry from Operation IceBridge, as well as seven publicly released Arctic SIT products. The validation results demonstrate that our SIT product has accuracy equivalent to existing products. The



accuracy of our products varies from 0.2 m to 0.4 m according to the input satellite

altimetry data. The SIT datasets are available on the Zenodo at https://doi.org/10.5281/zenodo. 13699698 (Xiao et al., 2024).

## 1    Introduction

Arctic sea ice is a crucial component of the Earth's climate system. Its growth and

melting affect regional and global climates through interactions with the atmosphere and ocean. Sea ice and overlying snow have a higher albedo than seawater, reflecting more solar radiation to the atmosphere, thereby modulating the ocean's energy budget. The ice-albedo positive feedback plays a key role in the interaction between sea ice and climate. Sea ice also acts as a barrier that restricts heat loss from a warm ocean to a cold

atmosphere. Moreover, the seasonal freeze and thaw of sea ice affect sea surface salinity, which in turn influences ocean currents. Sea ice is also an indicator of climate change. In the past decades, warming in the Arctic has occurred at four times the rate of the global average, a phenomenon referred to as Arctic amplification (Serreze et al., 2009; Rantanen et al., 2022). Arctic sea ice extent has declined rapidly in recent decades,

particularly since 2000. This decrease has also accelerated, leading to several occurrences of a record low Arctic sea ice extent in recent years. A recent study revealed that the Arctic could become ice-free in less than a decade even in the lowest-emission scenarios (Kim et al., 2023). Therefore, the observation and monitoring of Arctic sea ice is of great significance to accurately understand the influence and response of Arctic

sea ice on the global system.

Sea ice is characterized by areal extent, concentration, thickness, movement, and age, among other factors. Thickness is an important indicator of sea ice because it adds a third dimension to ice cover. Sea ice thickness (SIT) is altered by both thermodynamic and dynamic processes (von Albedyll et al., 2022). Its changes are important in

modulating the heat flux between the ocean and the atmosphere (Hall, 2004). Precise knowledge of SIT is crucial for interpreting the current summer and winter sea ice

decline (Stroeve and Notz, 2018) and for improving predictions of sea ice loss in the 21st century using climate models (Massonnet et al., 2018).

While satellite sensors have provided data on sea ice extent and concentration since 1979, long-term observations of SIT remain limited as ice thickness cannot be measured directly by satellite-based instruments. In addition to satellite remote sensing, SIT can be measured through field surveys, which include in situ drilling, underway measurements, airborne laser altimetry, and upward-looking sonar (ULS). However, due to the limited spatial sampling of field surveys, these SIT observations may not accurately represent basin-scale conditions. Moreover, field surveys are constrained by weather conditions, making continuous observations difficult. Therefore, satellite remote sensing, including satellite altimetry and passive microwave (PMW) technology, is a valid method for SIT observation, especially for obtaining hemispherical SIT.

SIT can be estimated with brightness temperature data from PMW sensors, such as the Soil Moisture and Ocean Salinity (SMOS) sensor. This estimation is based on an empirical relationship between the emissivity of ice and its thickness. PMW sensors have a larger footprint, allowing them to generate daily SIT estimates for the entire Arctic region. However, PMW-based SIT estimation is limited to first-year or thin ice, as PMW observations typically cannot penetrate ice thicker than 50 cm. This method is not directly sensitive to thicker sea ice, especially at higher frequencies (Heygster et al., 2014).

Alternatively, satellite altimetry has become an important technique for obtaining comprehensive hemispherical SIT. The method, first introduced by Laxon et al. (2003), involves estimating the ice freeboard—the height of the ice above sea level—by measuring the elevation difference between the sea ice and nearby leads. Using parameters such as freeboard loading snow depth and density of the ice, the SIT is then calculated under the assumption of hydrostatic equilibrium. Snow depth is an important factor limiting the accuracy of SIT estimates, as uncertainty in snow depth can account for up to 70% of the total uncertainty in the SIT estimate (Zygmuntowska et al., 2014).



A widely used snow depth dataset is the climatology of monthly snow depth (W99)
      created by Warren et al. (1999). W99 was generated with in situ snow depth
      observations from Soviet drifting stations between 1954 and 1991. However, since
      these observations were taken on multi-year ice (MYI), W99 does not accurately
      represent snow depth on first-year ice (FYI). Kurtz and Farrell (2011) demonstrated

that the mean snow depth from W99 over FYI differs significantly from airborne
      observations made during Operation IceBridge (OIB). In addition, given that the data
      are several decades old and were fitted using a two-dimensional quadratic function for
      each month independently of the year, W99 does not fully reflect current snow depth
      conditions, especially as the Arctic climate has undergone significant changes.

Furthermore, Webster et al. (2014) found that snow depth from W99 was overestimated
      by 37% on FYI and by more than 50% on MYI when compared with observations from
      OIB. Another way to obtain large-scale snow depth is through PMW sensors (Markus
      et al., 2006). However, PMW-based snow depth estimations are limited to dry snow
      within 50 cm thick as MYI and snow have similar effects on brightness temperatures

(Rostosky et al., 2018).

      Several Arctic SIT products are available from various institutions, including the Centre
      for Polar Observation and Modelling (CPOM), Alfred Wegener Institute's (AWI)
      Helmholtz Centre for Polar and Marine Research, National Snow and Ice Data Center
      (NSIDC), European Space Agency Climate Change Initiative (ESA CCI), and Centre

of Topography of the Oceans and the Hydrosphere (CTOH). However, these SIT
      products differ in spatial and temporal resolution, measurement uncertainties, and
      methods of estimation. For example, the spatial resolutions of different products vary
      from 5 km to 25 km. Currently, the longest time series dataset available for the Arctic
      SIT is from CTOH, which covers the period from 1994 to 2023; however, it has a coarse

resolution of 12.5 km. Although laser altimetry satellites feature small footprints and
      high single-point accuracy, they have limited temporal coverage. Radar altimetry
      satellites in polar orbits, by contrast, have been continuously monitoring Arctic sea ice



since the 1990s. Therefore, the main purpose of this study is to create an Arctic SIT

product for the period from 1995 to 2023 using multi-source satellite radar altimetry

data. The aim is to provide valuable data for improving research on the variations in

Arctic sea ice and its future evolution trends.

**Table 1 Acronyms and Abbreviations.**

| Acronyms | Meaning |
|---|---|
| ATM | airborne topographic mapper |
| AWI | Alfred Wegener Institute |
| BGEP | Beaufort Gyre Exploration Projec |
| C3S | Copernicus Climate Change Services |
| CCI | Climate Change Initiative |
| CDR | climate data records |
| CPOM | Centre for Polar Observation and Modelling |
| CryoVEx | CryoSat-2 Validation Experiment |
| CTOH | Centre of Topography of the Oceans and the Hydrosphere |
| DMS | digital mapping system |
| DTU | Technical University of Denmark |
| ERS | European Remote-Sensing Satellite |
| ESA | European Space Agency |
| FYI | first-year ice |
| GSFC | Goddard Space Flight Center |
| LEM | lowest elevation method |
| LSA | least squares adjustment |
| MAE | mean absolute error |
| MSS | mean sea surface |
| MWC | modified Warren climatology |
| MYI | multi-year ice |
| NERC | Natural Environment Research Council |
| NESOSIM | NASA Eulerian Snow on Sea Ice Model |
| NRT | near real-time |
| NSIDC | National Snow and Ice Data Center |
| OIB | Operation IceBridge |
| PMW | passive microwave |
| PP | pulse peakiness |
| RA-2 | Radar Altimeter 2 |
| REAPER | REprocessing of Altimeter Products for ERS |
| RMSD | root mean square deviation |
| RMSE | root mean squared error |
| SIRAL | SAR/Interferometric Radar ALtimeter |
| SIT | sea ice thickness |



| SMOS | Soil Moisture and Ocean Salinity |
| --- | --- |
| SSD | stack standard deviation |
| SSH | sea surface height |
| SSHA | sea surface height anomaly |
| STD | standard deviation |
| ULS | upward-looking sonar |
| W99 | Warren 99 Snow on Sea Ice Climatology |
| WHU SIT | Wuhan Unviersty Sea Ice Thickness |

## 2 Data

### 2.1 Satellite radar altimetry data

**2.1.1 ERS-2 data**

The European Remote-Sensing Satellite (ERS) program was the first initiative in Earth observation, aimed at providing comprehensive environmental monitoring. The mission detected land and ocean surface changes and provided observation data on oceans, polar ice, vegetation, geology, meteorology, and ecology. ERS-2, launched in

April 1995, was the second satellite of the ERS mission. ERS-2 operated on a 35-day repeat cycle, providing observations of Arctic sea ice cover south of 81.5°N. The satellite was decommissioned in July 2011. Here, we used ERS-2 products of version RP01 from the REAPER (REprocessing of Altimeter Products for ERS) project (Brockley et al., 2017).

**2.1.2 Envisat data**

Envisat, the successor to ERS, was launched in March 2002 and decommissioned in April 2012. The mission aimed to enhance the capabilities of the ERS program, particularly in ocean and ice monitoring. Envisat's orbital period was 35 days, the same as ERS-2. It was equipped with the Radar Altimeter 2 (RA-2), which is a dual-frequency,

nadir-pointing radar operating at 13.575 GHz (Ku-band) and 3.2 GHz (S-band). This radar measures the two-way delay of echoes from the Earth's surface with a high precision. We used the latest version 3.0 of the RA-2 product, which offers improved data coverage, validity, and quality at crossover points compared with previous datasets.

**2.1.3 CryoSat-2 Data**

CryoSat-2 is Europe's first ice mission to monitor the most dynamic sections of the Earth's cryosphere. With an advanced radar altimeter onboard, named SIRAL (SAR/Interferometric Radar ALtimeter), CryoSat-2 borrows synthetic aperture radar and interferometry techniques from standard imaging radar missions to sharpen its accuracy over rugged ice sheet margins and sea ice in polar waters. In addition to its

high accuracy, CryoSat-2 also features dense track spacing and a smaller data gap. The across-track spacing of CryoSat-2 is approximately 2.5 km at 75° and 4 km at 60°, which is a significant improvement compared with the coarse across-track spacing of 25 km at 75° and 4 km at 60° provided by ERS-2 and Envisat. The narrow across-track spacing allows for extensive data coverage along the edges of the Antarctic ice sheet

and Arctic sea ice. Operated in a non-sun-synchronous low Earth orbit at 92° inclination, CryoSat-2 provides data coverage up to 88°S/N, which is also a significant improvement over previous altimetry satellites. CryoSat-2 has a 369-day repeat cycle with a 30-day sub-cycle, which enables monthly coverage of Arctic sea ice. For this study, we used the latest Baseline E data from ESA, which includes significant

improvements to the CryoSat-2 ice products, such as improved sea surface height anomaly (SSHA) interpolation and advancements in land-ice retracking.

### 2.2 Publicly available satellite-based Arctic SIT products

Table 2 lists seven publicly available Arctic SIT products derived from satellite

altimetry and PMW data. These products were used to compare with our SIT product. It is important to note that these satellite-based SIT products are available only from October to April, as melt ponds on the sea ice during the Arctic summer months cause measurement issues.

**2.2.1 CPOM**

CPOM is part of the Natural Environment Research Council (NERC) that studies land ice, sea ice, and ice sheets using satellite observations and numerical models of the polar



regions. CPOM was the first to provide publicly available SIT estimates from CryoSat-2. CPOM provides monthly Arctic SIT gridded data at a resolution of 5 km and 1 km.

The 5 km resolution thickness data is available for the entire Arctic grid, while the 1 km resolution data is available for individual sectors. CPOM also offers near real-time (NRT) SIT products for 28, 14, and 2 d observation periods (Tilling et al., 2016). However, these NRT data do not include the precise orbit determination and atmospheric corrections found in the standard data.

The CPOM algorithm uses fixed criteria for stack standard deviation and pulse peakiness of CryoSat-2 waveforms to differentiate between leads and ice floes (Laxon et al., 2013). During the SIT calculation of CPOM, the depth and density of snow are based on the W99 dataset and are applied according to ice type. For MYI, the monthly mean climatological value is applied, and this value is halved for FYI based on

comparisons between W99 and OIB observations (Kurtz and Farrell, 2011). A dual ice density with 882 kg/m$^3$ for MYI and 917 kg/m$^3$ for FYI was used during the conversion of freeboard to thickness. The CPOM SIT was validated with airborne measurements from OIB and CryoSat-2 Validation Experiment (CryoVEx), as well as the ULS buoy observations in the Beaufort Gyre Exploration Project (BGEP), the standard deviations

between the CPOM and the three independent observations estimates are 66, 55, and 34 cm, respectively (Tilling et al., 2018).

### 2.2.2 AWI-CS2

The SIT product at the AWI began to be compiled from 2014 to evaluate the mass

balance of Arctic sea ice and the uncertainties associated with CryoSat-2 sea ice altimetry. The AWI-CS2 has been the basis for the climate data records (CDR) of European initiatives such as the ESA CCI and the Copernicus Climate Change Services (C3S). The AWI-CS2 provides monthly SIT datasets as well as other information including sea ice freeboard and concentration on a 25 km grid.



In the latest version of AWI-CS2 (V2.6), CryoSat-2 ICE baseline-E L1B data are used
       for the full data record. AWI-CS2 applies a 50% threshold first-maximum retracker
       algorithm to derive elevation data for all surface types (Ricker et al., 2014). For snow
       depth, an earlier version of the AWI-CS2 followed the method in CPOM-CS2, hereafter
       referred to as the "modified Warren climatology" (MWC). In version 2.1 and after, a

monthly snow depth and density parametrization based on merging of the W99
       climatology and daily snow depth over FYI from AMSR2 data was introduced. The
       AWI-CS2 shows systematic uncertainties of 0.6 m for FYI and 1.2 m for MYI (Ricker
       et al., 2014).

**2.2.3   AWI-CS2+SMOS**

       AWI-CS2+SMOS is a blended product that combines data from CryoSat-2 and SMOS.
       It provides weekly Arctic SIT at a grid resolution of 25 km (Ricker et al., 2017).
       Combining CryoSat-2 and SMOS enhances SIT information and increases the update
       rate of Arctic-wide maps. CryoSat-2 performs better over thick ice, while SMOS

provides accurate measurements of thin ice thickness. The combination of both datasets
       is based on a statistical approach (optimal interpolation) that merges weekly
       information from CryoSat-2 and SMOS from AWI based on the respective uncertainties
       for different thickness classes. The merged ice thickness is compared to airborne
       electromagnetic induction sounding measurements in the Barents Sea, and has a bias of

-0.1 m and a root mean square deviation (RMSD) of about 0.3 m (Ricker et al., 2017).

       **2.2.4   CCI**

       The objective of the CCI Sea Ice project is to improve the ability to retrieve data on
       polar sea ice and to establish a long-term record of two key variables: concentration and

thickness. The CCI data record is based on radar altimetry data from the Envisat (2002-
       2012) and CryoSat-2 mission (2010-2017). It is available on a monthly grid with a



resolution of 25 km × 25 km during the Arctic winter season. The MWC was also used in the conversation of freeboard to thickness.

CCI consists of primary input data from two different radar altimeter missions. Due to the different characteristics of the two altimeters, the Envisat-based SIT and CryoSat-2-based SIT are separated in the gridded products. Although the CCI algorithms tried to minimize the inter-mission bias, a residual bias remains: Envisat freeboards in MYI regions are thinner than CryoSat-2 freeboards, while Envisat provides thicker freeboards than CryoSat-2 in regions that are dominated by FYI (Paul et al., 2018). While comparing with airborne electromagnetic thickness, the CCI SIT shows a root mean square deviation (RMSD) of 0.73 m for CryoSat-2 estimates and a RMSD of 0.90 m for Envisat estimates.

### 2.2.5 CTOH

CTOH is a French Observation Service dedicated to developing altimetric products for the long-term monitoring of sea level and ocean currents, lake and river levels, the cryosphere, and the planet's climate. CTOH'SIT product is based on altimetry missions, including ERS-1, ERS-2, Envisat, and CryoSat-2 (Bocquet et al., 2023). Data are available as EASE2-grid monthly maps with a spatial resolution of 12.5 km covering the period 1994–2023. The dataset also includes radar freeboard, sea ice freeboard, total freeboard, and SIT, as well as external variables such as sea ice density and concentration.

The freeboard of CTOH is measured by one altimeter according to the method in Laxon et al. (2013), whereas the snow depth measurements are based on multiple sources. For the period from 2013 to 2019, snow depth was estimated using the bi-frequency altimetry data from the CryoSat-2 Ku altimeter and the Saral Ka altimeter (Guerreiro et al., 2016). For other periods, MWC was used for snow depth estimation. The comparison of CTOH SIT with ULS measurements in BGEP shows a RMSE of 12-28 cm for Envisat period and 15-21 cm for CryoSat-2 period (Guerreiro et al., 2017).




### 2.2.6 GSFC-CS2

Another CryoSat-2-based SIT product is the GSFC-CS2, released by Kurtz and Harbeck (2017) from the Goddard Space Flight Center (GSFC). The SIT data are provided daily on a 25 km grid as 30-day averages for the months between September

and May. In the generation of GSFC-CS2, the sea ice elevation is first determined using a physical model to determine the best fit to each CryoSat-2 waveform (Kurtz et al., 2014). Sea ice freeboard is then determined by subtracting the gridded sea surface elevation from the gridded sea ice floe elevation and applying the radar propagation speed correction where snow depth data is available. Snow depth is also constructed

from modified W99, the same as CPOM. A key difference of the GSFC-CS2 product compared with other CryoSat-2 thickness products is that it uses a single ice density value of 915 kg/m³ for all ice types. The GSFC-CS2 shows a mean bias of 0.182 m with OIB observations (Kurtz et al., 2014).

### 2.2.7 GSFC-IS2

GSFC-IS2 was created using ICESat-2 ATL10 along-track sea ice freeboards and the NASA Eulerian Snow on Sea Ice Model (NESOSIM, Petty et al., 2023). ATL10 contains along-track estimates of sea ice freeboard using height and surface type information from ATL07. NESOSIM is a three-dimensional, two-layer (vertical),

Eulerian snow on sea ice budget model developed with the primary aim of generating daily estimates of the depth and density of snow on sea ice across the polar oceans (Petty et al., 2020). The monthly thickness data of GSFC-IS2 are binned to a 25 km × 25 km polar stereographic north grid. The GSFC-IS2 estimates shows a mean bias of 0.11 ± 0.20 with BGEP ULSs (Petty et al., 2023).


**Table 2. Information of publicly available satellite-based Arctic SIT products.**

| Product name | Source data | Temporal resolution | Space resolution | Period | Spatial coverage | Reference |
| --- | --- | --- | --- | --- | --- | --- |



| | | | | | | |
|---|---|---|---|---|---|---|
| AWI-CS2 | CryoSat-2 | Monthly | 25 km | 2010 to present | North of 60°N | Hendricks and Ricker (2020) |
| AWI- CS2+SMOS | CryoSat-2, SMOS | weekly | 25 km | 2011 to present | North of 60°N | Ricker et al. (2017) |
| CCI | Envisat, CryoSat-2 | Monthly | 25 km | 2002-2017 | North of 16.6°N | Hendricks (2018) |
| CPOM | CryoSat-2 | Monthly | 5–25 km | 2010 to present | North of 60°N | Laxon et al. (2013) |
| CTOH | ERS-1/2, Envisat, CryoSat-2 | Monthly | 12.5 km | 1994-2023 | North of 65°N | Bocquet et al. (2023) |
| GSFC-CS2 | CryoSat-2 | 30-days | 25 km | 2010 to present | North of 55°N | Kurtz and Harbeck (2017) |
| GSFC-IS2 | ICESat-2 | Monthly | 25 km | 2018-2022 | North of 60°N | Petty et al. (2022) |

### 2.3 Validation data

In addition to comparing our results with existing SIT products, we also used sea ice

draft data from ULS and airborne altimetry observations to validate our estimates. As

shown in Figure 1, the sea ice draft data are from four ULSs mounted as part of    BGEP

by the Woods Hole Oceanographic Institution (Kemp et al., 2005). ULS A, B, and C

have been operating since August 2003, but ULS C was out of service in August 2008.

In August 2005, ULS D came into operation. The stated accuracy of each acoustic range

measurement is +/- 5 cm, and the estimate error of the ice draft measurements after

implementing the processing procedure is +/- 5-10 cm

The airborne observations are from OIB, which was launched to bridge the observation

gap between ICESat and ICESat-2. The airborne topographic mapper (ATM) and snow

radar are combined to derive the sea ice freeboard and thickness. The flight lines of OIB

over sea ice from 2009 to 2019 are illustrated in Figure 1. The IceBridge L4 and Quick

Look Sea Ice Freeboard, Snow Depth, and Thickness products (Kurtz et al., 2015) were

used for validation. The freeboard uncertainty of OIB is 5 cm approximately (Yi et al.,

2015), with an uncertainty of 5 cm of snow radar observation (Webster et al., 2014).





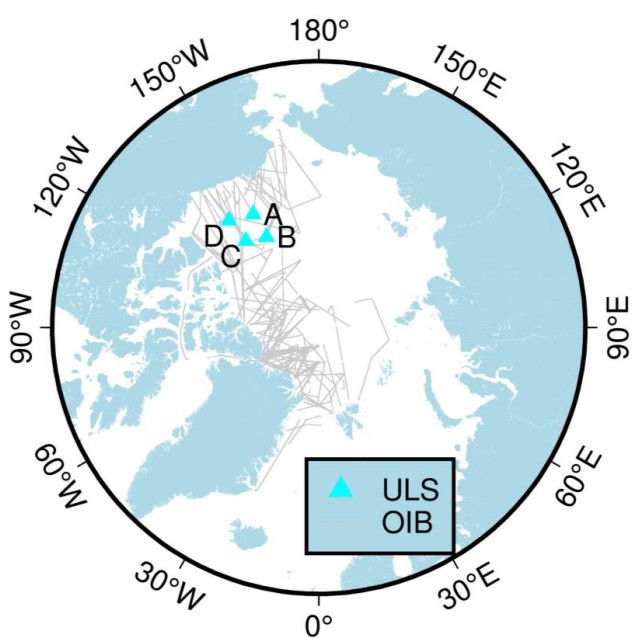

**Figure 1. Locations of the four ULSs (light-blue triangles) and flight lines of the**

**OIB sea ice observations (gray lines).**

### 2.4 Auxiliary data

Table 3 lists the auxiliary datasets used for the sea ice freeboard and thickness

calculations. DTU18MSS provides global high-resolution mean sea surface (MSS) data

and is released by the Technical University of Denmark (DTU, Andersen et al., 2018).

The major advance of DTU18MSS to previous versions is the usage of 3 years of

Sentinel-3A data and an enhanced 7-year record from Cryosat-2.

The Sea Ice Concentrations dataset from Nimbus-7 SMMR and DMSP SSM/I-SSMIS

Passive Microwave Data (NSIDC-0051) provide a consistent time series of sea ice

concentrations using multiple PMW data sources from October 1978 to the present

(DiGirolamo et al., 2022). The data are posted in the polar stereographic projection at

a grid cell size of 25 km. Here, we define ice floe regions as those with a sea ice

concentration greater than 75%.



EASE-Grid Sea Ice Age (NSIDC-0611) provides weekly estimates of sea ice age from

1984 to 2022 for the Arctic Ocean based on remotely sensed sea ice motion and sea ice

extent (Tschudi et al., 2019a). The data are projected using a 12.5 km Northern

Hemisphere EASE-Grid. Recent sea ice age data can be accessed in the quicklook

version of the EASE-Grid Sea Ice Age product (NSIDC-0749, Tschudi et al., 2019b).

The W99 climatology was compiled from measured field snow depth data from drifting

Soviet ice stations in 1937 and 1954–1991 (Warren et al., 1999). It provides monthly

snow depth and density data with a 2-dimensional quadratic function. This dataset was

used to convert the sea ice draft from ULSs to SIT.

**Table 3. Information of auxiliary datasets**

| Datasets | Parameter | Temporal coverage | Temporal resolution | Spatial coverage | Spatial resolution | Usage |
|---|---|---|---|---|---|---|
| DTU18MSS | Mean sea surface height | 1998–2018 | Static | Global | 1 minute | Freeboard retrieval |
| NSIDC-0051 | Sea ice concentration | 1978–2023 | 1 day, 1 month | Global | 25 km | Ice floe regions define |
| NSIDC-0611/0749 | Sea ice age | 1984–2022 | 7 day | Northern Hemisphere | 12.5 km | Sea ice type define |
| W99 Climatology | Snow depth | \ | Monthly | Northern Hemisphere | \ | Draft to thickness |

## 3    Methods

### 3.1 Sea ice freeboard retrieval

Altimetric SIT is calculated from the freeboard and other parameters with the

assumption of hydrostatic equilibrium. Sea ice freeboard is determined as the height of

sea ice above water and can be calculated from the difference between sea ice elevation

and sea surface height (SSH). The sea ice elevation can be determined directly from the

altimeter, whereas the SSH is derived from leads nearby. Leads are linear-like areas





with open water or thin ice within the sea ice cover. Therefore, leads detection is crucial

for freeboard retrieval.

A popular method to identify leads from ice floes is to analyze the waveform shape since radar echoes reflected from leads are specular and echoes from ice are diffuse (Peacock and Laxon, 2004). For ERS-2 and Envisat, the pulse peakiness (PP), defined as the ratio of maximum to mean return power of the waveform (Tilling et al., 2018), is

used for leads identification. For CryoSat-2, stack standard deviation (SSD), provided in the L1b product, is also used for leads identification. The thresholds for leads and ice floes classification are summarized in Table 4. Echoes out of this classification criterion were removed as ambiguous.

**Table 4. PP and SSD thresholds for leads and floes classification.**

| Mission | Leads | Floes |
|---------|-------|-------|
| ERS-2 | PP > 30 | PP < 3 |
| Envisat | PP > 30 | PP < 3 |
| CryoSat-2 | PP > 18 and SSD < 4 | PP < 9 and SSD > 4 |

Figure 2(a) shows an example of the results of leads identification using waveform thresholds for CryoSat-2. The blue points are ice floes, while the yellow points indicate leads. We compared the identification results with images from OIB digital mapping

system (DMS). The sampling time difference between CryoSat-2 and OIB was within 2 hours. In general, as shown in Figure 2(e) and (g), the waveform threshold method can provide accurate identification results. However, in thin ice-covered areas such as shown in Figure 2(c) and (f), misidentification occurs. This is because specular echoes also occur when the radar burst is reflected from thin ice, such as grease ice/nilas and

newly frozen leads. If these echoes are misidentified as leads, an overestimation will occur on the SSH determination, thereby leading to an underestimation of the ice freeboard.

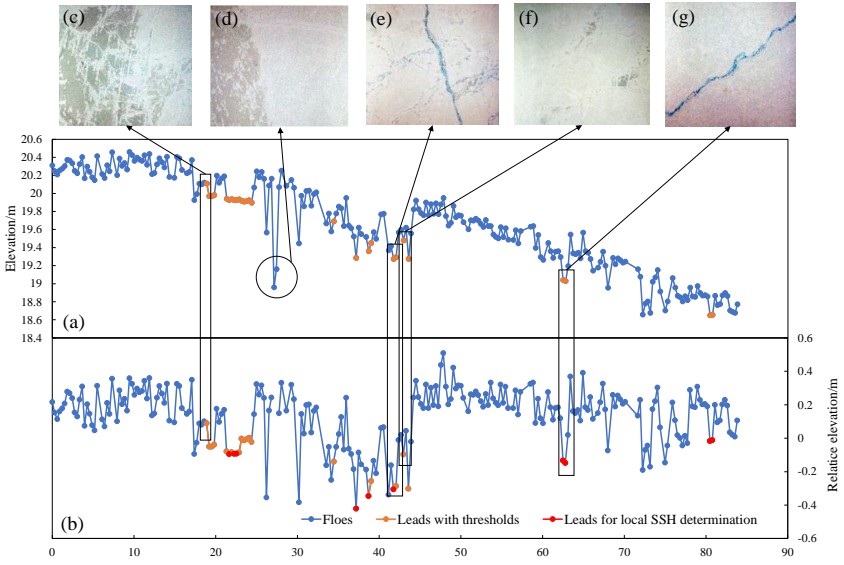

**Figure 2. Elevation (a) and relative elevation (b) profiles of CryoSat-2 tracks. The**

**blue points indicate floes, and the yellow points indicate leads identified using the**

**threshold method; the red points indicate leads used for local SSH determination.**

**(c)–(g) OIB DMS images.**

To prevent the misidentification of leads when using waveform thresholds, we

introduced the lowest elevation method (LEM). LEM is based on the premise that the

surface height of leads is theoretically lower than that of the nearby sea ice surface.

LEM has been applied in Arctic sea ice freeboard retrieval using Envisat (Zhang et al.,

2021) and OIB (Zhang et al., 2022) altimetry data. To achieve accurate leads detection,

we combined LEM with the waveform threshold method. First, we calculated the

surface relative elevation by subtracting the MSS height, which was obtained from the

DTU18 MSS model. Geoid undulations were removed in the relative elevation. We then

applied statistical discrimination to the relative elevation profile based on the Pauta

criterion. Relative elevations beyond $h_{25km} \pm 3 \cdot STD$ were regarded as outliers, where

$h_{25km}$ and $STD$ are the mean and standard deviation (STD) of the relative elevation for

a 25-km section along the track, respectively. Figure 2(b) shows the corrected relative

elevation profile. Two outliers marked with a circle in Figure 2(a) were removed. For a certain 25-km section with more than three points identified as leads with the threshold method, the local SSH was determined by averaging the three lowest relative elevations. For sections with two or more points identified as leads, the local SSH was calculated

directly from the average relative elevation of the identified leads. For sections without identified leads, the local SSH was interpolated from adjacent sections. The red points in Figure 2(b) denote leads used for local SSH determination.

Finally, the sea ice freeboard was calculated by subtracting the local SSH from the relative elevation. We compared our freeboard estimates with those from the CryoSat-

2 Baseline E product and the spatiotemporally coincident OIB freeboard. The comparison is shown in Figure 3. The OIB freeboard was modified with snow depth from the snow radar. The mean freeboard along this track in this study was approximately 0.274 m, while the mean freeboard from the Baseline E product was 0.238 m. The mean value of the modified OIB freeboard was 0.261. Our freeboard

estimates were closer to the modified OIB freeboard than the freeboard in the Baseline E product. As mentioned previously, the waveform threshold method leads to an underestimation of the freeboard, which explains why the freeboard in the Baseline E product was smaller than our estimates and the modified OIB freeboard.

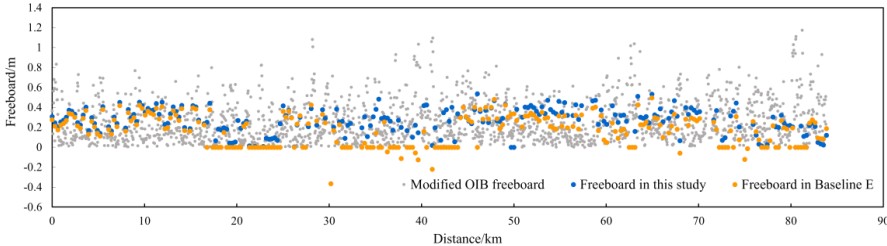


**Figure 3. A comparison of the freeboard results from this study, the CryoSat-2 Baseline E product, and the OIB. The gray points indicate the modified OIB freeboard, the blue points indicate the freeboard estimate in this study, and the yellow points indicate the freeboard results in the CryoSat-2 Baseline E product.**






## 3.2 Conversion of freeboard to thickness

The retrieved ice freeboard can be converted to SIT by assuming hydrostatic equilibrium. The conversion involves parameters including the densities of sea ice, seawater, and onloading snow, as well as snow depth. However, these parameters are difficult to obtain concurrently with altimeter measurements, which introduces uncertainties in the thickness calculations. Following the study of Xiao et al. (2020), we proposed a method based on least squares adjustment (LSA) to convert CryoSat-2 freeboard data to SIT. A quadratic model (Equation 1) between freeboard and thickness was established within a 5 km × 5 km grid, and the SIT was calculated based on the LSA method:

$$
\begin{aligned}
h_{fb}(x, y) &= (1 - \frac{\rho_{si}}{\rho_{sw}})\overline{h_{si}} + (1 - \frac{\rho_s}{\rho_{sw}} - \theta)h_s + a_0 x + a_1 y + a_2 x^2 + a_3 y^2 + a_4 xy \\
&= a_0 x + a_1 y + a_2 x^2 + a_3 y^2 + a_4 xy + a_5 \overline{h_{si}} + a_6
\end{aligned}
\tag{1}
$$

where $h_{fb}$ is the ice freeboard, $\overline{h_{si}}$ is the mean ice thickness of the grid, $h_s$ is the snow depth, $\theta$ is the penetration factor of radar signals, $a_0 - a_6$ are coefficients of the model, and $\rho_{sw}$, $\rho_{si}$, and $\rho_s$ are the densities of seawater, sea ice, and snow, respectively. $x$ and $y$ represent the longitudinal and latitudinal surface distances between the observation point and the central point of the grid cell.

To determine the SIT as well as the model coefficients using the LSA method, at least eight observations are needed in each grid. Figure 4 shows the numbers of ERS-2, Envisat, and CryoSat-2 observations falling within each 5 km grid cell. The percentage of grids with more than eight observations for ERS-2, Envisat, and CryoSat-2 was 63.88%, 73.38%, and 76.15%, respectively. The calculation was repeated using a 25 km × 25 km grid. Missing values in the 5 km grid were filled by resampling data from the 10 km grid.

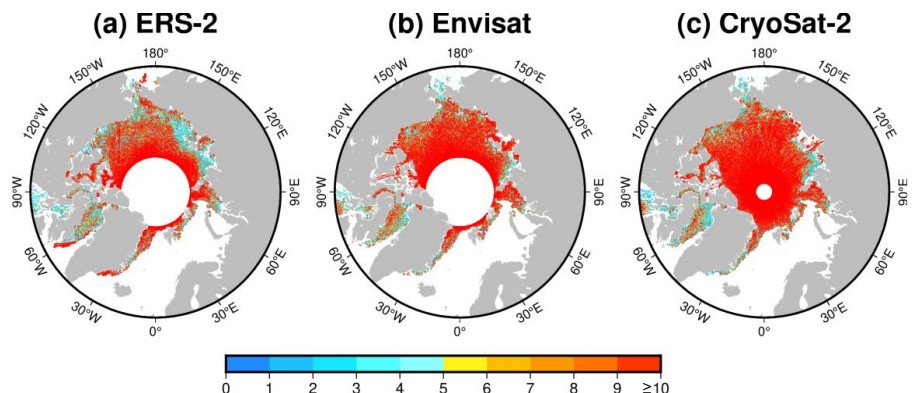


**Figure 4. Numbers of ERS-2, Envisat, and CryoSat-2 observations falling within each 5 km grid cell.**

### 3.3 Inter-mission consistency

Our SIT results involve three different radar altimetry missions. The radar altimeters on ERS-2 and Envisat are pulse-limited antimeres, while the SIRAL on CryoSat-2 uses SAR beam sharpening. The pulse-limited altimeters have a large footprint of 2–10 km over sea ice. The large footprint is more susceptible to specular returns and hence increased mixing of different surface types. Unlike the pulse-limited altimeters, SIRAL

features a much smaller footprint of 1.65 km × 0.3 km, owing to the combination of SAR technology and Doppler post-processing.

CryoSat-2 observations during the common mission period were used to correct the Envisat-based freeboard and thickness (Guerreiro et al., 2017; Paul et al., 2018; Tilling et al., 2019; Bocquet et al., 2023). Significant relationships were found between the

Envisat waveform parameters and the freeboard discrepancy between CryoSat-2 and Envisat. Guerreiro et al. (2017) built a fitting equation to correct the Envisat freeboard based on the relationship between the freeboard differences and the PP of Envisat's waveforms. Paul et al. (2018) developed an adaptive retracker threshold approach that uses differences in freeboard, surface backscatter, and the leading-edge width of

Envisat's waveforms. The retracker approach was applied to Envisat's waveforms to





minimize the freeboard biases. Bocquest et al. (2023) presented a multiparameter neural-network-based method for calibrating freeboard measurements from Envisat and ERS-2. Tilling et al. (2019) developed a physical-based approach to correct Envisat SIT according to the relationship between the thickness differences between Envisat and

CryoSat-2 and the along-track distance between leads and the closest floe in the Envisat measurements.

We first calculated SIT using Envisat and CryoSat-2 data based on the method in section 3.1. As shown in Figure 5, there were visible differences in the two thickness distributions, especially in areas of the Baffin Bay and the Canadian Arctic Archipelago.

Figure 6 shows a histogram of SIT from Envisat and CryoSat-2. The mean thickness from Envisat was 1.65 m, while the mean CryoSat-2-based thickness south of 81.5°N was 1.46 m. Compared with CryoSat-2 thickness, Envisat thickness showed an overestimation of $0.19 \pm 0.67$ m in January 2011.

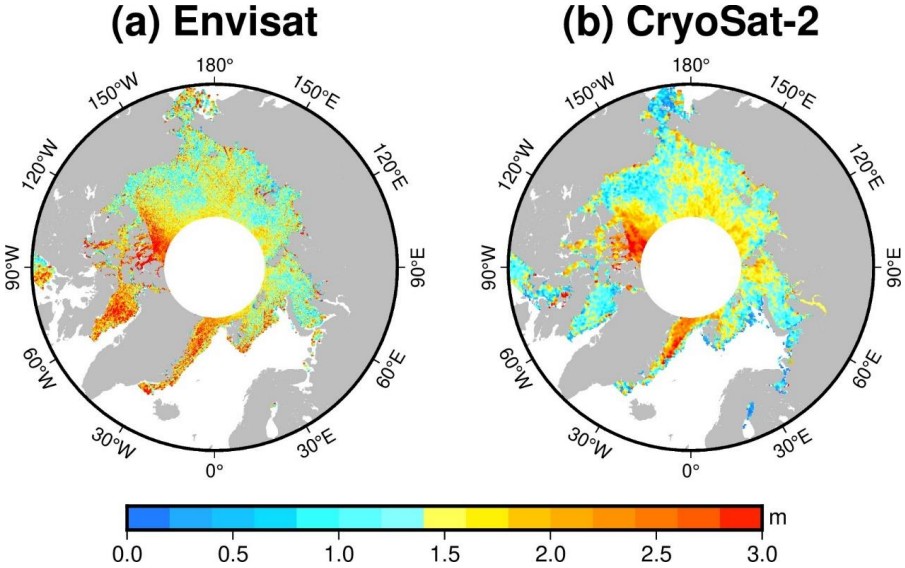

**Figure 5. Sea ice thickness from Envisat (a) and CryoSat-2 (b) for January 2011. The CryoSat-2 thickness is limited to 81.5°N.**

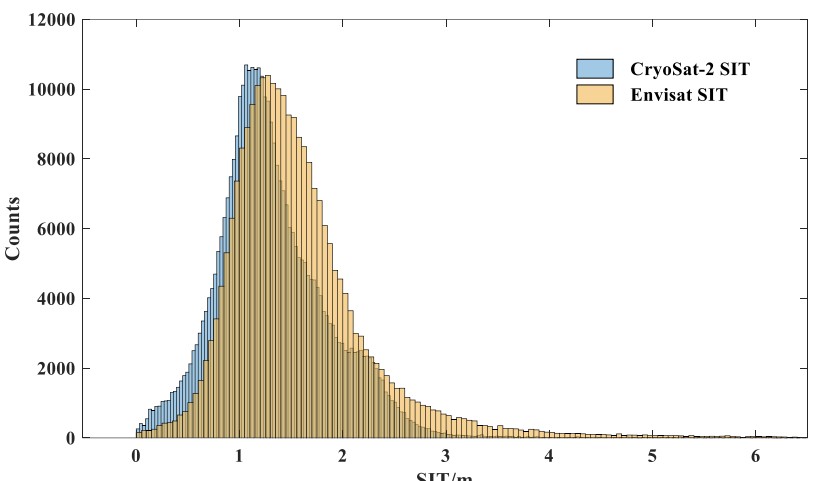

**Figure 6. Histogram of sea ice thickness in January 2011 from Envisat (in yellow) and CryoSat-2 (in blue).**

To minimize the inter-mission bias, we first compared the Envisat and CryoSat-2 thickness during the common mission period within a 5 km grid. Table 5 shows the monthly average difference and STDs between Envisat and CryoSat-2 thickness during the common mission period. From January to April, the mean difference was approximately 0.2 m, while from October to December, the difference was larger. We then generated correction grids for each month by averaging the differences within the grid cell. Finally, the monthly correction grids were applied to the Envisat thickness of the corresponding months. Figure 7(a) shows the modified Envisat SIT. Significant corrections can be observed in Baffin Bay and the Canadian Arctic Archipelago. The bias between the modified Envisat thickness and CryoSat-2 thickness was approximately 0.05 ± 0.37 m.

**Table 5. Statistics of the difference between Envisat and CryoSat-2 thickness during the common mission period.**

| Month | 2010 | | 2011 | | 2012 | |
|---|---|---|---|---|---|---|
| | Mean | STD | Mean | STD | Mean | STD |
| 1 | | | 0.19 | 0.67 | 0.22 | 0.65 |





| | | | | | | |
|---|---|---|---|---|---|---|
| 2 | | | 0.12 | 0.68 | 0.17 | 0.68 |
| 3 | | | 0.14 | 0.70 | 0.12 | 0.68 |
| 4 | | | 0.15 | 0.72 | | |
| 10 | | | 0.34 | 0.57 | | |
| 11 | 0.31 | 0.61 | 0.40 | 0.62 | | |
| 12 | 0.33 | 0.63 | 0.35 | 0.63 | | |


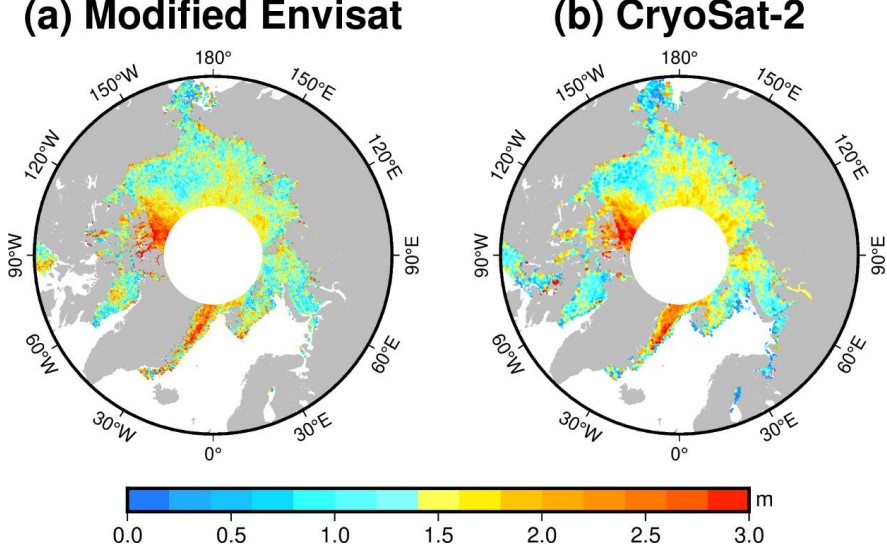

**Figure 7. Modified Envisat sea ice thickness (a) and CryoSat-2 thickness (b) for January 2011. The CryoSat-2 thickness is limited to 81.5°N.**

Figure 8 shows the difference in thickness between ERS-2 and Envisat during the common period in April 2003. This difference is approximately −0.39 m and is negligible compared with the difference between CryoSat-2 and Envisat since the altimeters on ERS-2 and Envisat are similar. Thus, we also applied the monthly correction gird to the ERS-2-based thickness for correction.





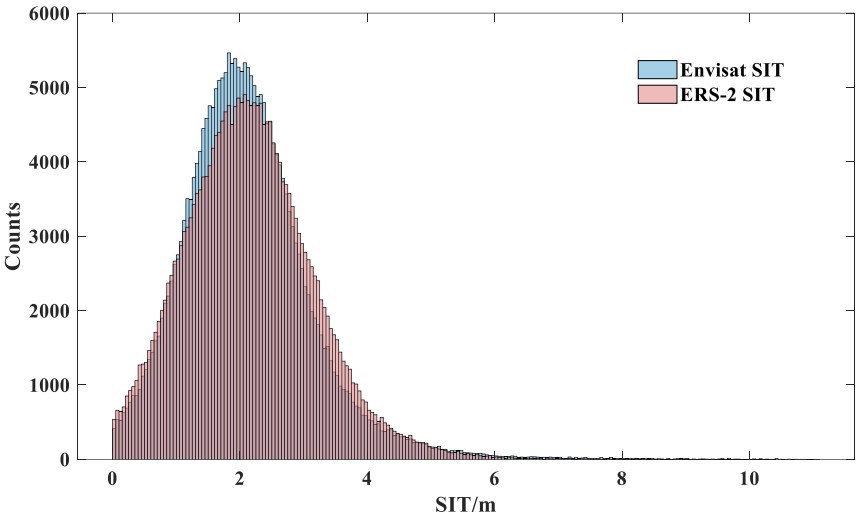


**Figure 8. Histogram of sea ice thickness in April 2003 from Envisat (in brown) and ERS-2 (in kermesinus).**

## 4   Results

**4.1 Sea ice thickness distributions**

We finally developed an Arctic SIT product for the period from 1995 to 2023 using altimetry data from ERS-2, Envisat, and CryoSat-2. The product is hereafter named WHU SIT (Wuhan Unviersty Sea Ice Thickness). Figure 9 shows the spatial distributions of Arctic SIT from WHU SIT during 2023. Due to the existence of melt

ponds in melt seasons, ice thickness for May to September is not provided. Figure 10 shows statistic histograms of ice thickness for 2023. The mean thickness is 1.59 m in January, growing to a maximum of 1.94 m in April. The sea ice extent did not show any significant changes during this growth. In January, a small area of thick ice (larger than 2.5 m) can be seen in the northeast of Greenland and north of the Canadian Arctic

Archipelago. With continuous ice freezing, more thick ice appeared. After summer melting, the mean SIT decreased to a minimum of 1.13 m in October. The sea ice extent was also the smallest in October. Seasonal sea ice melted away during summer and new





ice did not yet form in the marginal seas. SIT and extent continued to grow from October to December.

As shown in Figure 10, the thickness distributions are close to a normal distribution. The median value of the histogram increased from January to April and from October to December. In April, the median value was approximately 2 m, while in October the median was only 1 m. We can find humps on the right side of the histograms of January to April. These humps indicate thick ice formed in these months. The histogram of

October shows a slight sinistrality, indicating that thin ice predominated at the start of the freezing season.

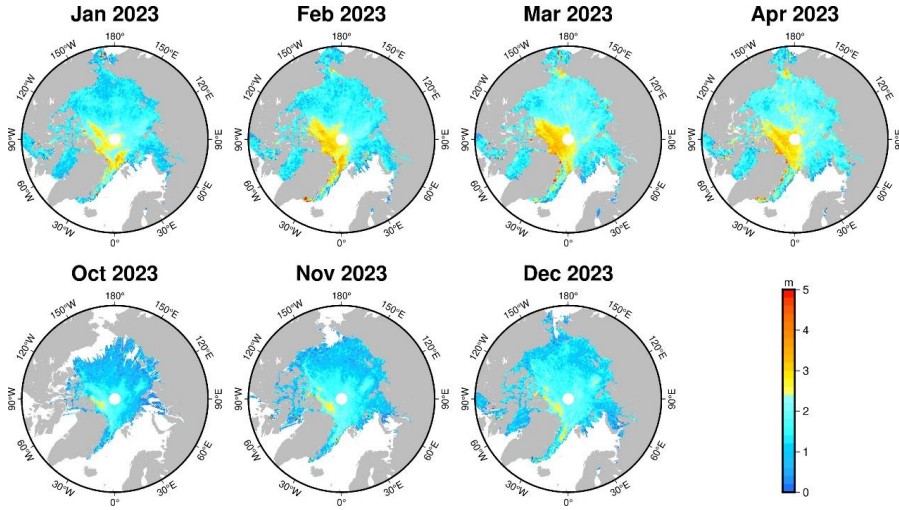

**Figure 9. Spatial distributions of Arctic sea ice thickness from WHU SIT in 2023.**

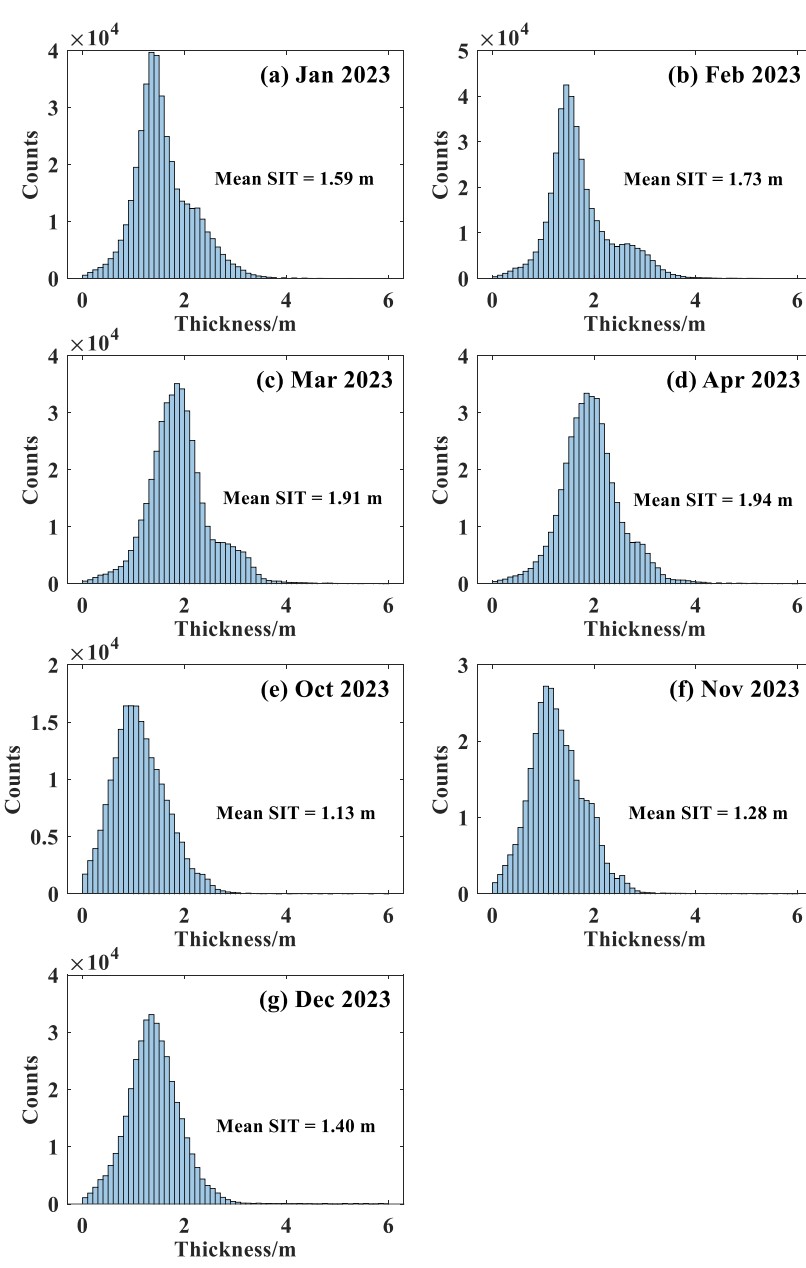


**Figure 10. Histograms of Arctic sea ice thickness from WHU SIT in 2023.**

**4.2 Variations in Arctic sea ice thickness**



Figure 11 shows the annual average thickness from WHU SIT south of 81.5°N from 1995 to 2023. The annual average SIT refers to the average thickness during the frozen season, namely from October to April of the next year. The annual SIT decreased from 1.72 m in 1995/1996 to 1.40 m in 2022/2023, with a decreasing rate of 0.014 m/yr. The largest annual average SIT during the past 28 years was 1.91 m in 1996/1997, while the smallest value was 1.34 m in 2012/2013. The STD of the annual average FYI, MYI, and total SIT in the 28 years was 0.06 m, 0.17 m, and 0.14 m, respectively, indicating that Arctic SIT variation is primarily driven by the variation in MYI. From 1997/1998 to 2006/2007, the SIT variation was relatively stable with a decreasing rate of 0.002 m/yr. In 2007/2008, the annual average thickness reached a historical low of 1.54 m. After that, the SIT increased by 0.03 m/yr till 2009/2010. In the following 3 years, the SIT decreased rapidly by 0.08 m/yr, reaching a new historical minimum of 1.34 m in 2012/2013. In 2013/2014, the SIT saw a significant increase of 0.23 m. From 2013/2014 to 2022/2023, the SIT decreased at 0.02 m/yr, though there were fluctuations throughout this period.

The variation in MYI thickness was close to that of the total SIT. The minimum thickness of MYI also occurred in 2012/2013. The mean MYI thickness decreased by 0.017 m/yr during the research period, while the variation in mean FYI thickness was much more moderate with a decreasing rate of 0.005 m/yr.

Figure 12 shows the annual average SIT south of 88°N in the Arctic from 2010 to 2023. The annual average SITs south of 88°N were larger than those south of 81.5°N. The differences in FYI, MYI, and total SIT between the area south of 88°N and south of 81.5°N were 0.08 m, 0.22 m, and 0.18 m, respectively. This is because MYI dominates in the region between 81.5°N and 88°N. However, the variations in SIT south of 88°N were close to the variations south of 81.5°N.

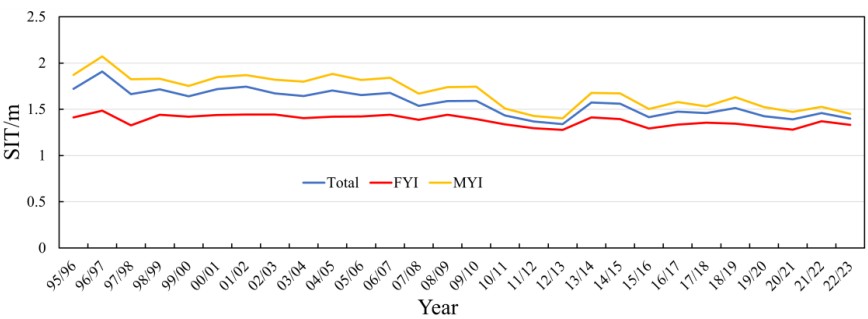

**Figure 11. Annual average thickness from WHU SIT south of 81.5°N in the Arctic from 1995 to 2023. The blue line indicates the total SIT, the red line indicates FYI, and the orange line indicates MYI.**

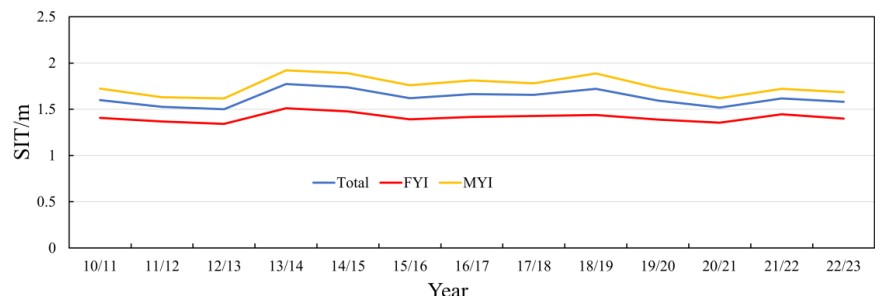

**Figure 12. Annual average thickness from WHU SIT south of 88°N in the Arctic from 2010 to 2023. The blue line indicates the total SIT, the red line indicates FYI, and the orange line indicates MYI.**

The variations in monthly average SIT from 1995 to 2023 are presented in Figure 13. On the whole, the monthly average SITs during the freezing season decreased from 1995 to 2023. The rate of decrease in MYI thickness was larger than that for FYI thickness. In October, the decreasing rates for FYI, MYI, and total SIT were 0.009 m/yr, 0.022 m/yr, and 0.021 m/yr, respectively. The largest decreasing rates for both FYI and MYI occurred in October during the freezing season. Conversely, the smallest decreasing rates for MYI and total sea ice occurred in January, while FYI showed its smallest decreasing rate in December.

The largest average thickness of total sea ice, FYI, and MYI for the seven months

mainly occurred in 1996 or 1997. The smallest average thickness of total sea ice and

MYI in October, November, and December occurred in 2011, and for the months of

January to April, the smallest values occurred in 2013. The lowest monthly average

thickness for FYI was more dispersed.

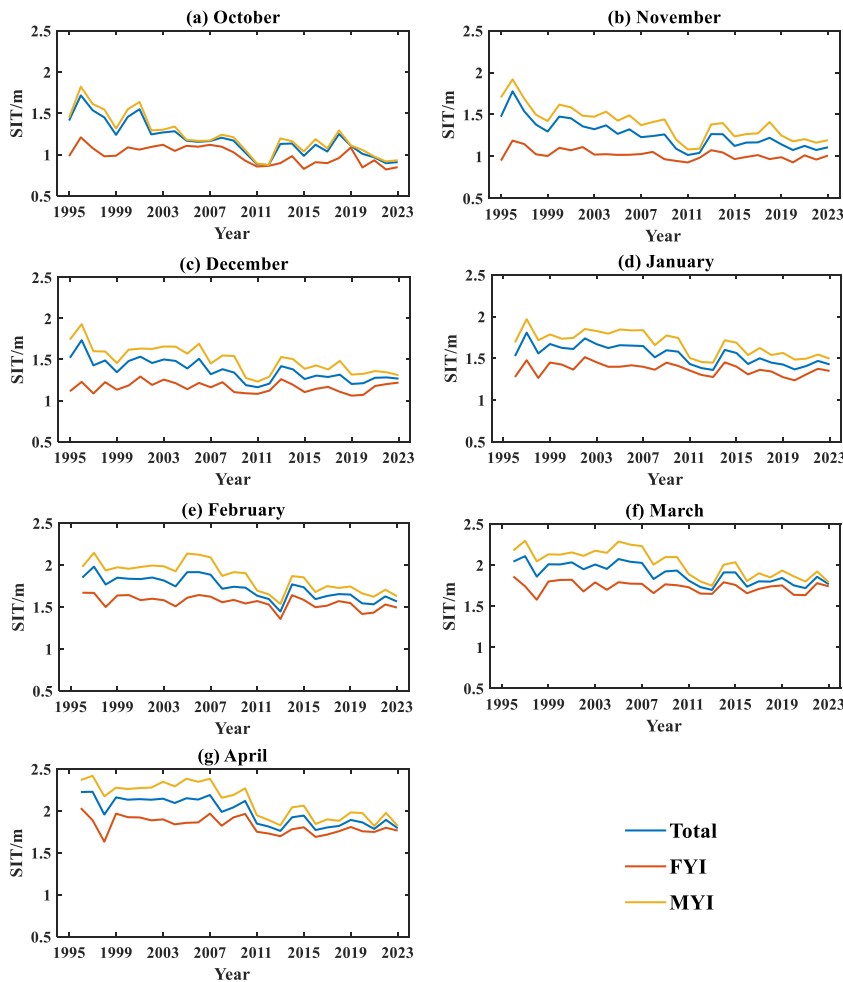

**Figure 13. Monthly average thickness from WHU SIT south of 81.5°N in the Arctic**

**from 1995 to 2023. The red line indicates the total Arctic sea ice, the blue line**

**indicates FYI, and the orange line indicates MYI.**

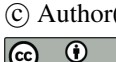

## 5 Validations

### 5.1 Comparison with other satellite-based products

We first compared WHU SIT with the seven SIT products listed in Table 2. Figure 14

shows the annual average SITs of the eight products from 1995 to 2023. Note that the

annual average SIT before October 2010 refers to the average SIT south of 81.5°N.

Table 5 shows the statistics of the differences in annual average SIT between WHU SIT

and the seven other products. As shown in Figure 14, WHU SIT estimates present

similar variations as the existing products. The similar variations can also be concluded

from the STDs in Table 6.

The WHU SIT shows moderate estimates among the eight products, while the GSFC-

CS2 shows the largest estimates, and the AWI-CS2+SMOS shows the lowest estimates.

The WHU SIT shows the largest mean absolute error (MAE) against GSFC-CS2 and

the smallest MAE against GSFC-IS2.


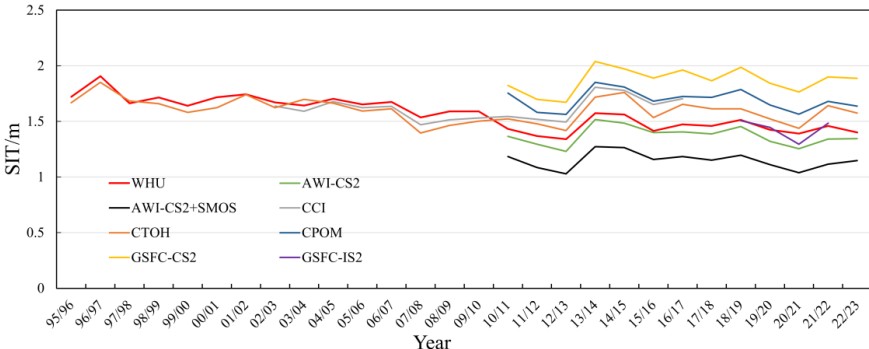

**Figure 14. Comparison of the annual average SITs from the eight products. The red line represents WHU in this study, the black line represents the merged CS2 and SMOS products from AWI, the orange line represents CTOH, the yellow line**

**represents GSFC-CS2, the green line represents AWI-CS2, the gray line represents CCI, the blue line represents CPOM, and the purple line represents GSFC-IS2.**





**Table 6. Statistics of the mean absolute error (MAE) and standard deviation (STD) between WHU SIT and the seven SIT products.**

| WHU- | AWI-CS2 | AWI-CS2+SMOS | CCI | CTOH | CPOM | GSFC-CS2 | GSFC-IS2 |
|---|---|---|---|---|---|---|---|
| MAE/m | 0.078 | 0.298 | 0.114 | 0.094 | 0.246 | 0.423 | 0.037 |
| STD/m | 0.032 | 0.032 | 0.128 | 0.104 | 0.037 | 0.055 | 0.056 |


### 5.2 Validation with ULS thickness

Figure 15 and Table 7–8 show the comparison results between ULS observations and the eight satellite-based SIT products. The draft data from ULS were converted to SIT using snow depth and density from W99. Generally, satellite-based SIT estimates 600 showed similar variations as observations from ULSs. Tables 7 and 8 present the statistics before and after October 2010, respectively. As shown in Table 7, our SIT estimates performed best in terms of MAE and STD at ULS B, C, and D. While at ULS A, our estimates were second to the CCI product. For the period after October 2010, the satellite-based SIT products performed better, mainly owing to the improvements 605 in CryoSat-2. The two products from AWI showed the lowest MAE and STD among the eight products. The STDs of WHU were close to those of CCI, CPOM, and CTOH, but WHU featured lower MAEs.

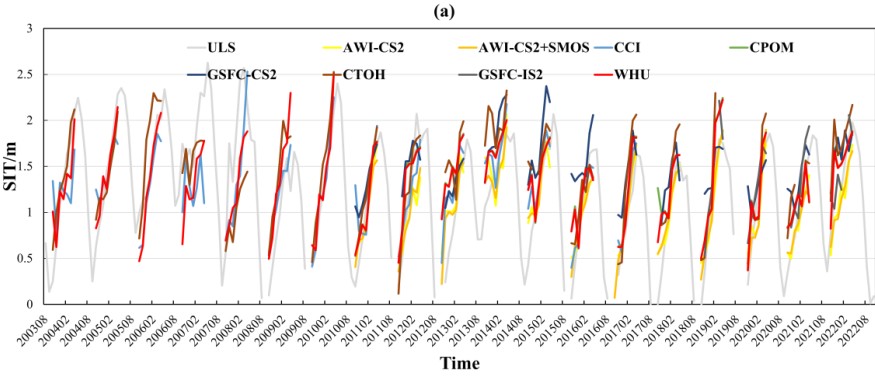



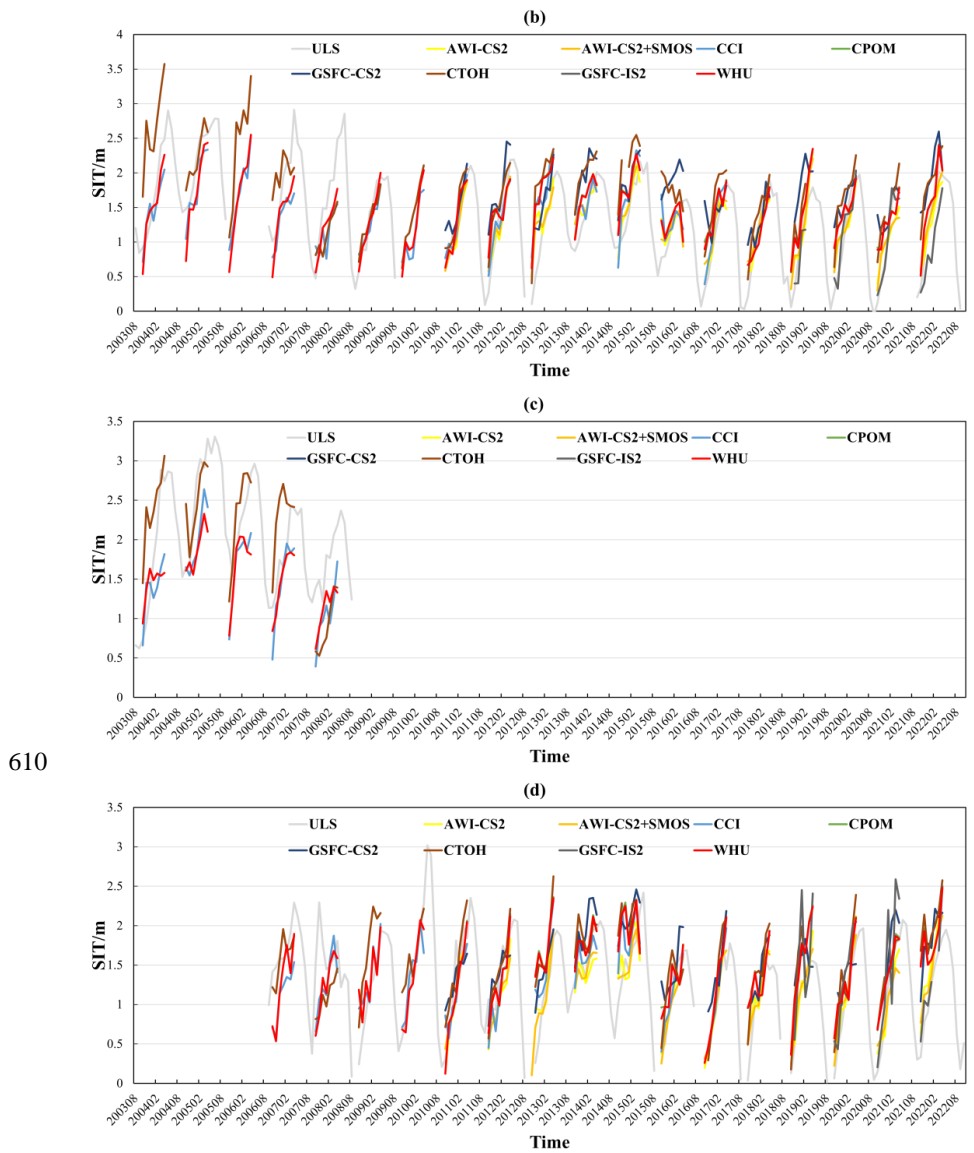

**Figure 15. Comparisons of SITs between the ULSs and the eight satellite-based products: (a) ULS A, (b) ULS B, (c) ULS C, and (d) ULS D. The locations of the four ULSs are shown in Figure 1.**


**Table 7. Statistics of MAE and STD between thickness measurements from the ULSs and the satellite-based products before October 2010.**



| | A | | B | | C | | D | |
|---|---|---|---|---|---|---|---|---|
| | MAE/m | STD/m | MAE/m | STD/m | MAE/m | STD/m | MAE/m | STD/m |
| CCI | 0.23 | 0.30 | 0.34 | 0.42 | 0.41 | 0.53 | 0.47 | 0.55 |
| CTOH | 0.31 | 0.42 | 0.65 | 0.67 | 0.88 | 0.84 | 0.54 | 0.67 |
| WHU | 0.26 | 0.35 | 0.31 | 0.39 | 0.38 | 0.47 | 0.43 | 0.51 |

**Table 8. Statistics of MAE and STD between thickness measurements from the ULSs and the satellite-based products after October 2010.**

| | A | | B | | D | |
|---|---|---|---|---|---|---|
| | MAE/m | STD/m | MAE/m | STD/m | MAE/m | STD/m |
| AWI-CS2 | 0.10 | 0.13 | 0.18 | 0.15 | 0.13 | 0.15 |
| AWI-CS2+SMOS | 0.09 | 0.12 | 0.19 | 0.13 | 0.10 | 0.13 |
| CCI | 0.18 | 0.12 | 0.33 | 0.22 | 0.28 | 0.25 |
| CPOM | 0.23 | 0.12 | 0.37 | 0.18 | 0.41 | 0.20 |
| CTOH | 0.40 | 0.11 | 0.59 | 0.16 | 0.48 | 0.18 |
| GSFC-CS2 | 0.32 | 0.18 | 0.58 | 0.20 | 0.45 | 0.20 |
| GSFC-IS2 | 0.22 | 0.13 | 0.18 | 0.25 | 0.21 | 0.26 |
| WHU | 0.21 | 0.12 | 0.15 | 0.18 | 0.16 | 0.23 |

**5.3 Validation with OIB thickness**

We also used the OIB thickness product for validation. The OIB observations were first allocated to the grids of the satellite-based products. Then, the mean thickness of the OIB within the grid was compared with the corresponding grid values.

Figure 16 and Table 9 show the histograms and statistics of the thickness differences between OIB and the satellite-based products. The differences between OIB and the seven satellite-based products showed similar distributions, exhibiting a normal distribution pattern; most of the differences fell between −1 m and 1 m. The mean SIT differences between CCI, CPOM, GSFC-CS2 and WHU against OIB were within 0.1 m, while for the other three products, the mean SIT differences were over 0.18 m. The MAEs and STDs between the satellite-based products and OIB SIT were close, indicating that the seven products have similar accuracy under the validation of OIB.



Specifically, our product had an MAE of 0.37 m and an STD of 0.43 m, presenting a

moderate accuracy among the seven products.

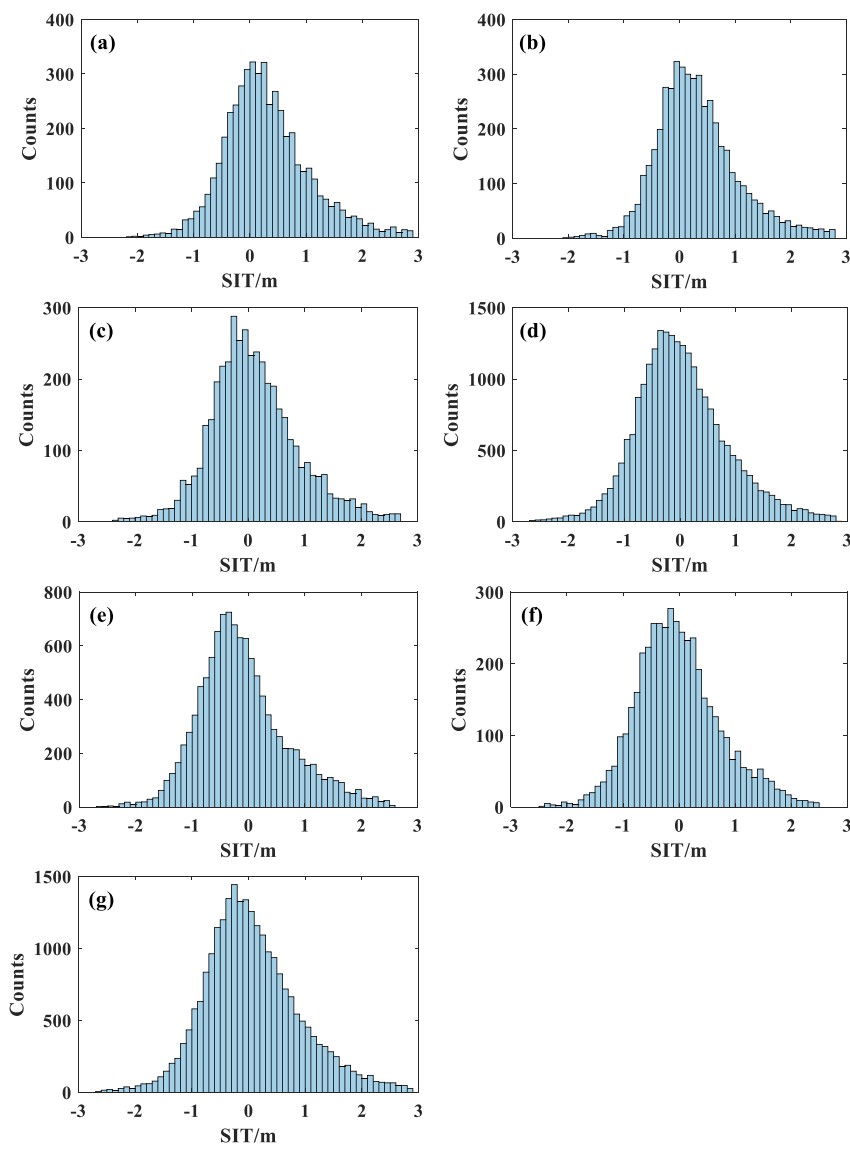

**Figure 16. Histograms of the thickness difference between OIB and the satellite-based products: (a) AWI-CS2, (b) AWI-CS2+SMOS, (c) CCI, (d) CPOM, (e) CTOH, (f) GSFC-CS2, and (g) WHU.**




**Table 9. Statistics of mean difference, MAE and STD between thickness measurements from OIB and the satellite-based products.**

|  | Mean/m | MAE/m | STD/m |
|---|---|---|---|
| AWI-CS2 | 0.24 | 0.39 | 0.42 |
| AWI-CS2+SMOS | 0.25 | 0.38 | 0.41 |
| CCI | 0.05 | 0.36 | 0.43 |
| CPOM | −0.02 | 0.39 | 0.47 |
| CTOH | −0.18 | 0.40 | 0.43 |
| GSFC-CS2 | −0.06 | 0.36 | 0.42 |
| WHU | 0.02 | 0.37 | 0.43 |

### 5.4 Uncertainties in altimetric SIT retrieval

When using the classic method to calculate thickness from freeboard under the assumption of hydrostatic equilibrium, the uncertainty of SIT can be computed as the error propagation of the input uncertainties including freeboard, ice density, snow depth and snow density. However, due to the lack of detailed observations of snow depth, snow density and sea ice density, it is difficult to evaluate the extent to which their

variability impacts on the retrieval accuracy. The uncertainties of these parameters are usually taken to be a constant or from empirical models. This will inevitably result in a distortion while assessing the uncertainties of altimetric SIT.

The SIT in this study is calculated with the quadratic model using LSA method. In LSA, the iteration stops when the variance of unit weight is stable. Thus, the uncertainties of

the SIT can be calculated by the difference of $\overline{h_{si}}$ in the last two iterations. Figure 17 shows the sea ice thickness uncertainty distribution with the LSA method for December 2023. The estimation uncertainty is less than 0.1 m in most areas of the Arctic Ocean. Large uncertainty can be found near the coast and sea ice margins.

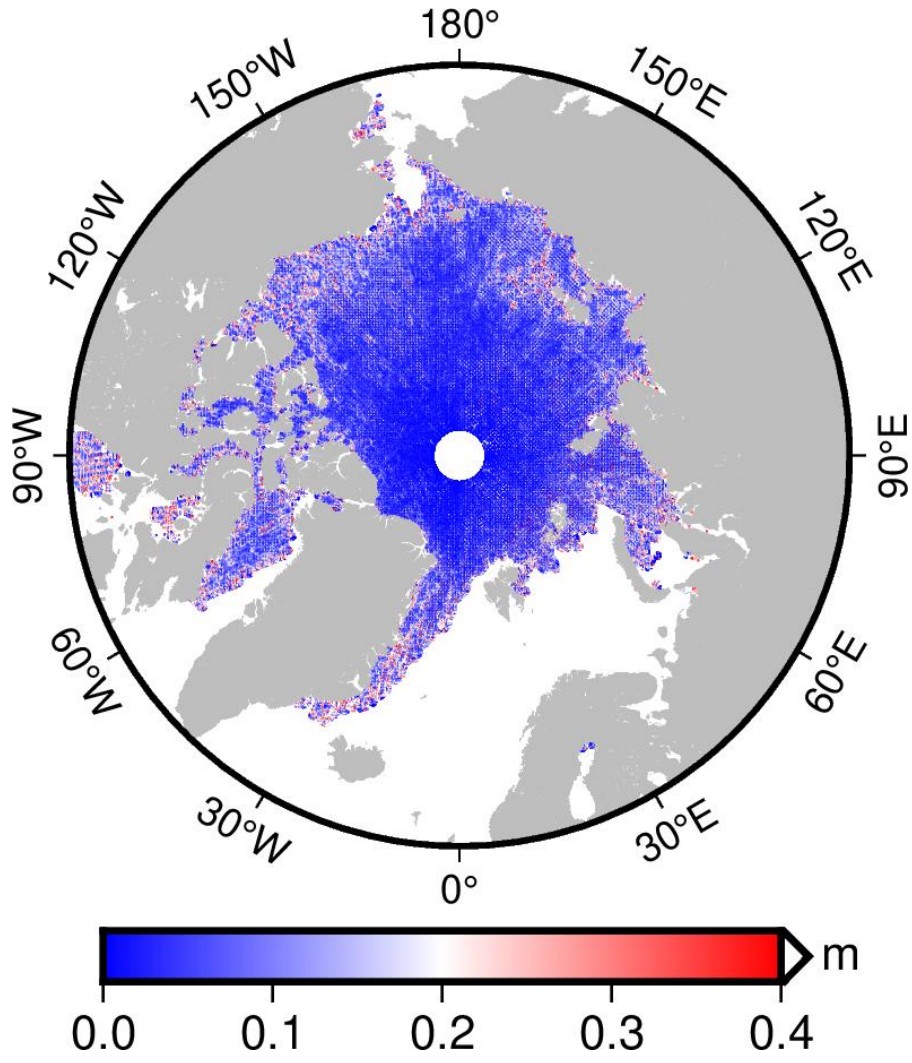


Figure 17. Arctic sea ice thickness uncertainty with the LSA method for December

2023.

## 6    Data availability

The    WHU    SIT    product    can    be    downloaded    from    Zenodo    at
https://zenodo.org/records/13699698 (Xiao et al., 2024). The datasets are provided in

NetCDF    (.nc;    Network    Common    Data    Form)    format    and    named



Arctic_sea_ice_thickness_5km_yyyymm_v1.0.mat, where yyyy and mm represent the year and month, respectively. We also provide maps of the thickness distributions in
PNG format.

ERS-2 GDR datasets can be found at ra-ftp-ds.eo.esa.int (last accessed on May 29, 2024). Envisat RA-2 GDR datasets can be found at ra2-ftp-ds.eo.esa.int (last accessed on May 29, 2024). CryoSat-2 GDR datasets can be found at science-pds.cryosat.esa.int (last accessed on May 29, 2024). AWI-CS2 and AWI-CS2+SMOS products can be
found at ftp.awi.de (last accessed on May 29, 2024). The CPOM product can be found at http://www.cpom.ucl.ac.uk/csopr/seaice.php (last accessed on May 29, 2024). The CCI product can be found at anon-ftp.ceda.ac.uk (last accessed on May 29, 2024). The CTOH product can be found at https://www.legos.omp.eu/ctoh/fr/produits-ctoh/ (last accessed on May 29, 2024). The GSFC-CS2 product can be found at
https://nsidc.org/data/rdeft4/versions/ (last accessed on May 29, 2024). The GSFC-IS2 product can be found at https://nsidc.org/data/is2sitmogr4/versions/2 (last accessed on May 29, 2024). ULS data can be found at https://www2.whoi.edu/site/beaufortgyre/ (last accessed on May 29, 2024). The OIB L4 and quick look SIT datasets can be found at https://nsidc.org/data/idcsi4/versions/1 and https://nsidc.org/data/nsidc-
0708/versions/1 (last accessed on May 29, 2024). Sea Ice Concentrations (NSIDC-0051) can be found at https://nsidc.org/data/nsidc-0051/versions/2 (last accessed on May 29, 2024). Sea ice age (NSIDC-0611 and NSIDC-0749) can be found at https://nsidc.org/data/nsidc-0611/versions/4 and https://nsidc.org/data/nsidc-0749/versions/1 (last accessed on May 29, 2024). DTU18MSS can be found at
ftp.space.dtu.dk/pub/Altimetry/FAMOS (last accessed on May 29, 2024).

## 7   Conclusions

In this study, we developed a new Arctic SIT product for the period from 1995 to 2023 by combining multiple radar altimetry data from ERS-2, Envisat, and CryoSat-2. The
SIT is presented on a monthly 5 km grid. We first improved the lead detection method

by combining the utilization of waveform parameter thresholds and lowest elevation. The improved method can eliminate the effects of grease ice, nilas, and newly frozen leads. The freeboard was then converted to thickness using a quadratic model based on hydrostatic equilibrium and least squares adjustment. We also generated a monthly

thickness correction grid using the common period observations of Envisat and CryoSat-2 to correct the inter-mission bias. The thickness difference between Envisat and CryoSat-2 was reduced from 0.67 m to 0.37 m after applying the correction grid. Finally, we generated a long time series of Arctic SIT estimates, along with their spatial and temporal distributions, from 1995 to 2023. The annual SIT decreased at a rate of

0.014 m/yr during the period. The decrease in rate was largest in October, reaching 0.021 m/yr. We compared our SIT product with seven publicly released products, ULS draft data, and OIB airborne observations. Our SIT estimates show similar variations as compared to existing products. The accuracy of our products is approximately 0.4 m when validated against ULS draft data from the Envisat period and approximately 0.18

m during the CryoSat-2 period. When validated against OIB observations, the accuracy of WHU SIT is approximately 0.02±0.43 m. In general, the newly developed SIT product shows good performance in terms of time series, spatial resolution, and accuracy compared with existing products.

**Author contribution**: FX performed the calculation and wrote the manuscript. SZ and FL contributed to the conception of the study. JL, TG, TL and HL contributed to discussions and analysis of the results. All authors contributed to improvement of the manuscript.

**Competing interests**: The authors declare that they have no conflict of interest.

**Acknowledgements**: We would like to thank the organizations that shared their datasets and software for use in this study.



**Financial support**: This research was supported by the National Key Research and Development Program of China under grant number 2023YFC2809103, the Fundamental Research Funds for the Central Universities under grant numbers 2042022kf1204, 2042022kf1069, 2042023gf0012, 2042022dx0001, the Hubei Provincial Natural Science Foundation of China under grant number 2022CFB081 and

the State Key Laboratory of Geodesy and Earth's Dynamics, Innovation Academy for Precision Measurement Science and Technology under grant number SKLGED2023-2-6.

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
