# Peer review of "A monthly Arctic sea ice thickness product from 1995 to"

_Earth System Science Data, 2024_

## Referee Comment (RC2)

Paper Review

**A monthly Arctic sea ice thickness product from 1995 to 2023 using multiple radar altimetry data**

Feng Xiao[1,2,3], Shengkai Zhang[1,2], Jiaxing Li[1,2], Tong Geng[1,2], Tingguo Lu[1,2], Hui Luo[1,2], Fei Li[1,2]

**General comment**

This paper presents a new time series of Arctic sea ice thickness over nearly 30 years based on altimetry observations with an unprecedented resolution of 5km. This is an important subject because only one other series of this length exists to date, and it may help to confirm the initial results and the relevance of this type of monitoring of the state of the sea ice. The comparison is even more interesting as the method used is new and totally independent of the previous ones. It relies on the multiplication of altimetric freeboard measurements in each 5km x 5km grid cell, to determine the thickness of the ice without preconceived ideas about parameters such as density or snow depth. For these reasons, I believe that this work deserves to be published.

Nevertheless, some important concepts have been overlooked, leading to dubious interpretations of certain results. In particular, the authors do not distinguish between the freeboard measured by Ku-band radar altimeters and the freeboard of the ice, which leads to erroneous equations that mainly affect the validation part. Because snow reduces the speed of propagation of the radar wave, the radar freeboard is necessarily smaller than the ice freeboard, and these 2 freeboards cannot be compared directly. The comparisons with OIB freeboard Figure 3 are therefore questionable. For the comparisons with ULS the used ice density is not specified. No correlation is calculated when comparing with other solutions or with in-situ measurements.

While ERS2 is a hard point because of the problem of blurring of its waveforms, no resulting map is shown. Although the data supplied with the paper seems to give consistent results, it would be useful to present comparisons with Envisat over a common month and to discuss any problems encountered, such as the extent of filtering used (Pauta criterion). Also, very strangely, the quality of the results deteriorates sharply, with extremely moth-eaten maps for the Envisat period from October 2002, even though the Envisat measurements are of much better quality than those of ERS2 and their orbits are the same. Coverage became good again with CryoSat-2 from November 2010, but visually the resolution seemed closer to 25km than 5km. This seems to indicate that you forgot to apply the coarse resolution to Envisat, but also that the 5km resolution is far too low to cover the Arctic basin.

All this raises questions about the choice of such a small resolution (5km), compensated by a relatively coarse resolution (25km). For a future study, wouldn't a resolution of

12.5km be more appropriate? Or do you think that 5km really adds value (relevant signal at this scale)?

Finally I would recommend analysing and exporting in the data the parameters a5 and a6, which depend on the densities and the snow depth. This should allow evaluating the consistency of these parameters, and thus the model used.

From a practical viewpoint I would recommend to use the Lambert-Azimuthal-Equal-Area projection with lon0=0, which becomes the reference in our domain, but it's not mandatory as it represents a lot of work without changing the results (see EASE v2 in https://nsidc.org/data/user-resources/help-center/guide-ease-grids). Strangely, while the figures in the paper show maps with lon0=0, the data are centred on lon0=-45.

Following all these considerations I recommend a major revision of this paper.

S.Fleury

**Detailed comments**

l.24 : The only mention of the projection used. Should be specify within the paper with all the requested parameters to use this projection, in particular the True Latitude (lat_ts). Also we would recommend to use the EASE v2 https://nsidc.org/data/user-resources/help-center/guide-ease-grids, which becomes the reference and which is much more convenient as the resulting grid is a regular Cartesian grid in meters and allows to compute directly distances, surfaces and volumes in ISU, which is not the case for the stereopolar projection.

l.46: please specify that the Arctic could become ice free in less than a decade in summer

l.95: I was surprised by these snow depth reduction announced over FYI and MYI but in fact in Webster et al. 2014, they do not speak about FYI/MYI but about specific seas: *"For the 2009–2013 period, the products show that snow has decreased by 37 ± 29% in the western Arctic and by 56 ± 33% in the Beaufort and Chukchi seas, compared to the 1954–1991 snow depth climatology produced by W99."* Please correct it or provide the original citation if any.

l.148: I suppose there is an error in this sentence: *"coarse across-track spacing of 25 km at 75° and 4 km at 60° provided by ERS-2 and Envisat"* as the across track spacing decreases with the latitude.

l.152: CryoSat-2 orbit is not any more a repeat cycle since several years, however there is still a sub-cycle of about 30 days.

l.185,248,270,285,286,293,294: It's very pertinent to recall the uncertainties for each product. I would just recommend to always use the same units (meters or centimetres).

l.195: CryoSat-2 ICE baseline-E L1B is the official ESA product, not an AWI product. Please specify if you use the AWI product or the ESA product.

l.238: This long time series is described in Bocquet et al. 2024  https://doi.org/10.

1029/2023JC020848 . Bocquet et al. 2023 explains the methodology and is only over Arctic from 1995 to 2021 (without ERS1).

l.249: "RMSE of 12-28 cm for Envisat period and 15-21 cm for CryoSat-2 period (Guerreiro et al., 2017)." Guerreiro or Bocquet? It's not the same time series.

l.300: DTU18MSS is endowed with discontinuity problems close to the MIZ and you should use DTU21MSS. See: "The DTU21 global mean sea surface and first evaluation" in Earth System Science Data 15(9):4065-4075 10.5194/essd-15-4065-2023

l.366: The MSS includes the geoid (MSS=geoid+MDT). Also I'm surprise that you don't correct for the usual altimetry corrections such as the wet tropo, the dry tropo, the ocean tide, load tide, pole tide, DAC, etc. You would get even more flat measurements.

l.367: I did not know about Pauta criterion, I thing it would be interesting to explain it here in few words.

l.375: *interpolated* is between 2 measures, here I would say *extrapolated.*

l.380-388: this part is not clear at all because you mixed-up the ice-freeboard (FBi, the real freeboard of the ice), the radar-ku-freeboard (FBku, the freeboard measured by the radar) and the total-freeboard (FBt, ice+snow freeboard that is measured by the lidar of OIB or IceSat-2). They are linked by the following relations: FBt = FBi + SD and FBku=FBi- ($c_v$/$c_s$ - 1) * SD, where SD is the Snow Depth and cv/cs is the ratio of the speed of light in vaccum and in snow. This ratio depends on the snow density $\rho_{snow}$. From Ulaby 1986 we have : $c_v$/$c_s$ =(1 + 0.00051 * $\rho_{snow}$)$^{1.5}$ *(Tiuri  et al. 1984 suggest:  $c_v$/$c_s$ = (1 + 1.7 $\rho_{snow}$ + 0.7 $\rho^2_{snow}$ )$^{0.5}$  ).* Each time you speak about freeboard you must specify which freeboard you are speaking about:

l.381: *"The OIB **total (or lidar)** freeboard was modified with snow depth".*  Here you should also specify if you have just removed the snow depth to get an **ice freeboard** or if you also have corrected for the speed propagation to get a **radar-ku-freeboard** for the following comparisons.

l.382: *"The mean **radar? ice? yours from OIB? yours from satellite?** freeboard along this track in this study"*

l.383: *"while the mean **radar** freeboard from the Baseline E"* .

l.384: *"The mean value of the modified OIB freeboard was 0.261"  -> The mean value of the **ice? radar?** freeboard obtained from OIB was 0.261m.*

l.387: The following sentence is wrong: *"the waveform threshold method leads to an underestimation of the freeboard ...".* The threshold method some times over estimates the FB and sometimes underestimates it as you show it later on in this paper. It mainly depends on the roughness of the ice, i.e. on ice type (FYI/MYI).

*"… which explains why the freeboard in the Baseline E product was smaller than our estimates and the modified OIB freeboard."* No, this is explained by the following equation FBku=FBi- ($c_v/c_s$ - 1)*SD, which shows that the **radar freeboard is always smaller than the ice freeboard** and it can even be negative for small FBice and large SD.

l.406: This equation is not the equation of the hydrostatic equilibrium between FBi, SIT and SD!

Here it is: FBi = SIT(1- $\rho_{ice}/ \rho_{water}$) - SD * $\rho_{snow} / \rho_{water}$

What you have written is the equation that links FBku with SIT and SD (ie, what you call $h_{fb}$ is in fact FBku and what you call theta is cv/cs including possibly a penetration factor P*cv/cs).

l.407-409: Please specify which are the inputs and which are the unknown.  It looks like that $h_{fb}$, x and y are the inputs and a0-a7 and $h_{si}$  are the 8 unknown, right?

l.410: x and y are really lat and lon? It's strange with your grid projection. Using EASE2 they could be directly in meters ;-)

l.418: I suppose it's 25km here, not *10km.* as you have just computed the 25km x 25km grids. However,, when you look at the data supplied, the maps appear very patchy. Can you explain this? (not enough measurements, even at 25km resolution?).

l.426: typo: *antimeres* -> altimeters

l.427*: "The pulse-limited altimeters have a large footprint of 2–10 km"* radius or diameter ?

l.441: typo: *Bocquest* -> Bocquet

l.442:  *"for calibrating freeboard measurements from Envisat and ERS-2."*  -> for calibrating Envisat freeboard measurements from CryoSat-2 and ERS-2 from calibrated Envisat.

l.443 : *"Tilling et al. (2019) developed a physical-based approach to correct Envisat SIT …"* Could be confusing with the retracker physical-based approach and it's not more physical than considering the ice roughness as it is usually done, I would avoid this term.

l.453: I do not agree with this conclusion :*"Compared with CryoSat-2 thickness, Envisat thickness showed an overestimation of 0.19 ± 0.67 m in January 2011."* As it is shown in your maps and histograms, LRM gets thinner ice over thin ice and thicker over thick ice relatively to SAR as it was explained in Laforge et al 2021 https://doi.org/10.1016/j.asr.2020.02.001

l.473, Table 5: Is 'Mean' the Mean Bias ? Is STD the STD of the difference or of the product?  Would be very pertinent to add the Correlations.

l.480: The main problem with ERS-1 and ERS-2 is related to the blurring of their waveforms over sea ice. You don't mention it and it looks like that you don't have

applied specific correction for this problem. Any other calibration between ERS2 and Envisat, as for Envisat versus CryoSat2? As it is an important problem it would also be important to see the maps you obtain in front of Envisat map for the same period. And once again the correlation is an important criteria that should also be added.

l.484: typo: *gird* -> grid

l.487, caption Fig 8: what is kermesinus ?

l.497: *"The sea ice extent did not show any significant changes during this growth."* How do you determine the sea ice extent? It's from NSIDC-0051 with concentration>75%? Would worth to recall it here. It is a very important point as the mean SIT highly depends on it (if you consider or not the thin ice in MIZ).

l.505: what do you mean by "normal distribution"?

l.510: what do you mean by "sinistrality"?

l.535: *"The mean MYI thickness decreased by 0.017 m/yr during the research period,".* Please provide explicitly the period.

l.538-570: I'm not fully convinced of the interest of mean SIT for all the Arctic as this value mainly depends on the considered sea ice extent. For instance it can remains about no ice but only some remaining fast ice at the coast to obtain large mean SIT but it means nothing. To make this type of comparison meaningful you could for instance always consider the same mask (region) for each given month. Or an alternative would be to compute to total volume instead of the mean SIT. However I will not ask you to change this, but at least you should explain specifically how you define the mask, for instance do you always use the area provided by NSIDC-0051 with concentration>75% for each month of each year?

l.574: Please specify if you use the same mask for all the products. It's important to make them comparable.

l.582: The information of the mask is even more important for the CS2SMOS product because it can cover larger region as it also considers thin ice at the margins thanks to SMOS.

l.592, Table 6: The mean bias and the correlation should be added.

l.598: it's really important to know which ice density and which snow depth you have chosen to convert the draft to SIT. If these values are not coherent with the product you compare you will necessarily get higher differences for this product. Please also provide the used equation and the input parameters (mainly ice density, SD is based on WC or MWC?).

l.606: typo: *"The STDs of WHU were close …"* -> *"The STDs of WHU **are** close …"*

l.618-620, Tables 7 & 8: Please add  the Mean Bias and the Correlation in these tables.

Indeed, if there is a significant bias, both the MAE and the STD will be high, but if the correlation is good it will indicate that the tendencies are coherent, which is the most important point to study change rate. (Also it is not necessary to recall the units in each column ;-).

l.624: "*Then, the mean thickness of the OIB within the grid was compared with the corresponding grid values.*" OIB products do not include SIT, how do you compute it? Please provide the equation and the input parameters used from OIB data.

l.636, Figure 16: For some products the count reaches nearly 1500 and for others it is lower than 300. The shape of the histograms being similar, it means that the number of measurements from one product to another can differ by a factor 3. How can you explain it? Is it because of the resolution of the original product?

l.643, Table 9: Please add the correlations.

l.647: "*... error propagation of the input uncertainties including **radar** freeboard, ice density, snow depth ...*"

l.655: "*Thus, the uncertainties of the SIT can be calculated by the difference of hsi in the last two iterations.*". I don't understand why the last two iterations are more relevant than the previous ones. To me, this is more a reflection of the speed of convergence of the LSA than the uncertainties. Please justify this solution. For example, you could more naturally assess the distance between the model and the measurements by calculating the STD or MAE between the model and the measurements.

l.678: Please replace the link https://www.legos.omp.eu/ctoh/fr/produits-ctoh/ by a more direct one:  http://dx.doi.org/10.6096/ctoh_sit_2023_01

---

## Author Comment (AC1)

We sincerely thank the Reviewer for the constructive comments on the manuscript. The reviewer's insights and suggestions have been extremely helpful in improving the quality of our work. We have thoroughly considered each comment and have made the following responses and revisions. The amendments are marked in red in the revision.

Next, we respond point by point to the comments.

**General comments:**

1. It is currently difficult to identify how this dataset improves upon previous SIT datasets. The authors need to articulate these improvements more clearly.

**Response:** Thanks for your suggestion. In this study, we developed a novel Arctic SIT product by combining multiple radar altimetry data from ERS-2, Envisat, and CryoSat-2. The SIT is presented on a monthly 5 km grid, which is highest in satellite altimetry-based SIT products. The time series of our products covers from 1995 to 2023, ranking second only to that of CTOH.

In addition, we have also proposed an innovative data processing method including leads detection, freeboard conversion to thickness, and inter-mission bias correction. We first improved the lead detection method by combining the utilization of waveform parameter thresholds and lowest elevation. The improved method can eliminate the effects of grease ice, nilas, and newly frozen leads. The freeboard was then converted to thickness using a quadratic model based on hydrostatic equilibrium and least squares adjustment. We also generated a monthly thickness correction grid using the common period observations of Envisat and CryoSat-2 to correct the inter-mission bias. The thickness difference between Envisat and CryoSat-2 was reduced from 0.67 m to 0.37 m after applying the correction grid.

We have elaborated on these improvements in the conclusion section.

2. The language and structure of the paper require refinement to enhance overall readability.

**Response:** We have engaged a professional scientific editor who is proficient in

English to review the language of the paper. The editor has checked for grammar errors, improved sentence structures, and ensured the clarity of the writing. Additionally, we have re-organized the paper to enhance its logical flow.

3.  The figures require better visual design to improve clarity and aesthetics.

**Response:** The figures have been polished according to your specific comments below. And we have offered high-resolution figures in the WORD file.

**Specific comments:**

4.  Line 22: "Finally, the monthly SIT estimates for the Arctic Ocean from October 1995 to December 2023 are generated." This statement is unclear. You only obtained SIT results from October to April of the following year. I understand that it is challenging to extract sea ice thickness during the summer, but this statement is misleading.

**Response:** This statement is revised as: Finally, the monthly SIT estimates datasets for the Arctic Ocean are derived for the freezing period spanning from October 1995 to December 2023.

5.  Lines 85-100: What is the purpose of describing the progress in snow depth research?

**Response:** Snow depth is an important factor limiting the accuracy of SIT estimates, as uncertainty in snow depth can account for up to 70% of the total uncertainty in the SIT estimate. In previous researches, snow depth from W99 or PMW sensors were used in SIT calculation. However, these datasets have large uncertainties in snow depth estimation. Therefore, we here described the limitations in snow depth research to introduce why we utilized the LSA method to convert freeboard to thickness.

6.  Line 111: "They have limited temporal coverage", what is the temporal coverage?

**Response:** The ICESat satellite covered from 2003 to 2009 and the ICESat-2 operated from 2018.

7.  The authors produced a SIT product with a temporal resolution of one month and a spatial resolution of 5 km. Lines 101-116: The authors should explain the issues with current products in terms of spatiotemporal resolution and coverage.

**Response:** Thanks for your suggestion. The temporal coverage of SIT product is determined by the coverage of satellites. Most of the current SIT products provide data during the CryoSat-2 period because the ERS-2 and Envisat satellite are pulse-limited and have a larger footprint. The larger footprint is more susceptible to specular returns and hence increased mixing of different surface types. Consequently, these larger footprints pose significant challenges to leads identification and freeboard retrieval, and also affects the spatial resolution of SIT products.

Therefore, in this study, we improved the lead detection method by combining the utilization of waveform parameter thresholds and lowest elevation. The improved method can eliminate the effects of grease ice, nilas, and newly frozen leads.

We have added explanations regarding this issue.

8.  Line 138: A reference is required here.

**Response:** We have added the following reference:

Legresy, B., Papa, F., Remy, F., Vinay, G., Van Den Bosch, M., and Zanife, O. Z.: ENVISAT radar altimeter measurements over continental surfaces and ice caps using the ICE-2 retracking algorithm, Remote Sens Environ, 95, 150–163, https://doi.org/10.1016/J.RSE.2004.11.018, 2005.

9.  There is no textual reference to Table 1 in the manuscript. It should be moved to the appendix.

**Response:** Thanks for your suggestion, Table 1 has been moved to the appendix.

10. Table 2 should be moved to Line 164.

**Response:** Table 2 has been moved to the suggested place.

11. Figure 2 is unclear: Figures (c)-(g) should be placed below (b) according to the sequence; The connections between the boxes in (a) and (b) and figures (c)-(g) are unclear. I suggest adding color-coded borders for each subfigure, corresponding to the colored boxes and arrows in (a) and (b); The font size of the coordinates in (a) and (b) is too small, and the y-axis label in (b) is reversed.

**Response:** Thanks for your suggestion. Figure 2 has been updated. To prevent the y-axis labels of (a) and (b) from overlapping, the y-axis of (b) is placed on the right-hand side.

12. Figure 4 appears to be a screenshot rather than an original figure. The titles for figures (a)-(c) seem incomplete.

**Response:** In the PDF version, the resolution of the figure is damaged. We have provided the high-resolution figures in the WORD file. We have added explanations for (a)-(c) in the caption.

13. I understand that the primary focus of this paper is to publish a new SIT product. However, I hope you can provide explanations for sea ice variation phenomena in the results section. For instance, in Line 530, why was the average SIT in 2012/2013 the historical minimum? This could be explained by citing relevant literature.

**Response:** Thanks for your suggestion. We have added the following explanations:
Correspondingly, the Arctic sea ice cover reached a record minimum in 2012 for the satellite era. Cui et al. (2015) demonstrated that in 2007 and 2012, there was a higher surface air temperature and sea level pressure, which was accompanied by increased surface specific humidity and a higher sea surface temperature. As a result, the strengthened poleward wind was conducive to the melting of summer Arctic sea ice in various regions during those two years.

14. Figures 11-15: All these figures are line plots of SIT. Why do some include grids while others do not? Additionally, the x-axis title is "Year" in all cases, but some

display "22/23," while others use "2023." My suggestion is to either unify the format or explain the differences.

**Response:** Grids are included in Figure 14. For Figure 11, 12 and 14, we calculated the mean SIT during the frozen season, namely from October to April of the next year. So we displayed the x-axis as "22/23". In Figure 13, we show the monthly average thickness from 1995 to 2023. Therefore, the x-axis refers to the certain year.

15. Logically, validation should precede sea ice thickness analysis.

**Response:** The validation section has been moved forward.

16. Figure 14: The primary focus is the WHU dataset (red line), but it is currently unclear. I suggest: (1) Increasing the color contrast for the red line. (2) Using dashed lines for other datasets to make WHU stand out.

**Response:** Thanks for your suggestion, Figure 14 has been updated.

17. I cannot understand the statistical evaluation in Table 6: (1) MAE measures the average magnitude of absolute errors. (2) STD measures the variability or dispersion of the data, but what does it aim to express here? (3) ME (Mean Error) is crucial for assessing the direction of errors. (4) R (correlation coefficient) is also important for assessing the linear relationship. Therefore, both ME and R should be added to provide a more comprehensive assessment.

**Response:** Thanks for your suggestion. We have added the statistics of ME and R for the comparison. Our product has the highest correlation with CPOM, reaching 0.937. Its correlation with the products of AWI-CS2, GSFC-CS2 and AWI-CS2+SMOS also exceeds 0.9, while the correlation with CCI is the lowest.

18. Line 601: Why is October 2010 used as the dividing line for comparing two periods? Please provide justification for this choice.

**Response:** Before October 2010, the SIT was calculated with pulse-limited altimetry data from ERS-2 and Envisat. The pulse-limited altimeters have larger footprint than

CryoSat-2 and lower accuracy. Therefore, October 2010 was used as the dividing line for comparing two periods.

19. Figure 15: The x-axis labels are unclear. It is unnecessary to label every tick; you can increase the spacing between tick marks. The most important consideration is to clearly convey the information.

**Response:** Figure 15 has been updated.

20. I understand your intention with A-D in Table 7 and Table 8, but you need to explain this explicitly in the table caption to ensure clarity.

**Response:** Thanks for your suggestion. We have updated the Table 7 and 8 and the captions.

---

## Author Comment (AC2)

We sincerely thank the Reviewer for the constructive comments on the manuscript. The reviewer's insights and suggestions have been extremely helpful in improving the quality of our work. We have thoroughly considered each comment and have made the following responses and revisions. The amendments are marked in red in the revision.

Next, we respond point by point to the comments.

**General Comments**

This paper presents a new time series of Arctic sea ice thickness over nearly 30 years based on altimetry observations with an unprecedented resolution of 5km. This is an important subject because only one other series of this length exists to date, and it may help to confirm the initial results and the relevance of this type of monitoring of the state of the sea ice. The comparison is even more interesting as the method used is new and totally independent of the previous ones. It relies on the multiplication of altimetric freeboard measurements in each 5km x 5km grid cell, to determine the thickness of the ice without preconceived ideas about parameters such as density or snow depth. For these reasons, I believe that this work deserves to be published.

Nevertheless, some important concepts have been overlooked, leading to dubious interpretations of certain results. In particular, the authors do not distinguish between the freeboard measured by Ku-band radar altimeters and the freeboard of the ice, which leads to erroneous equations that mainly affect the validation part. Because snow reduces the speed of propagation of the radar wave, the radar freeboard is necessarily smaller than the ice freeboard, and these 2 freeboards cannot be compared directly. The comparisons with OIB freeboard Figure 3 are therefore questionable. For the comparisons with ULS the used ice density is not specified. No correlation is calculated when comparing with other solutions or within-situ measurements.

While ERS2 is a hard point because of the problem of blurring of its waveforms, no resulting map is shown. Although the data supplied with the paper seems to give consistent results, it would be useful to present comparisons with Envisat over a common month and to discuss any problems encountered, such as the extent of filtering used (Pauta criterion). Also, very strangely, the quality of the results deteriorates sharply, with extremely moth-eaten maps for the Envisat period from October 2002, even though the Envisat measurements are of much better quality than those of ERS2 and their orbits are the same. Coverage became good again with CryoSat-2 from November 2010, but visually the resolution seemed closer to 25km than 5km. This seems to indicate that you forgot to apply the coarse resolution to Envisat, but also that the 5km resolution is far too low to cover the

Arctic basin.

All this raises questions about the choice of such a small resolution (5km), compensated by a relatively coarse resolution (25km). For a future study, wouldn't a resolution of 12.5km be more appropriate? Or do you think that 5km really adds value (relevant signal at this scale)?

Finally I would recommend analysing and exporting in the data the parameters a5 and a6, which depend on the densities and the snow depth. This should allow evaluating the consistency of these parameters, and thus the model used.

From a practical viewpoint I would recommend to use the Lambert-Azimuthal-Equal-Area projection with lon0=0, which becomes the reference in our domain, but it's not mandatory as it represents a lot of work without changing the results (see EASE v2 in https://nsidc.org/data/user-resources/help-center/guide-ease-grids). Strangely, while the figures in the paper show maps with lon0=0, the data are centred on lon0=-45.

**Response:** We appreciate your recognition of the significance of our work and the potential it holds for the study of Arctic sea ice thickness. We will address your concerns one by one:

**(1) Concepts on freeboard**

As shown in the following figure, we define the terminology of freeboard:

- ice freeboard ($FB_i$): refers to the elevation of the snow–ice interface above the local sea level;

- total freeboard ($FB_t$): refers to the elevation of the air–snow interface above the local sea level, which is sensed by laser altimetry;

- radar freeboard: as the radar waves do not fully penetrate snow above ice, we here define the term radar freeboard as the elevation of penetration interface above the local sea level (Ricker et al., 2014).

As for the ice freeboard the lower wave propagation speed in the snow layer requires a correction, the radar-ku-freeboard ($FB_{ku}$) is defined, but is not applied for the radar freeboard in this study. Therefore, the freeboard mentioned in this study refers to radar freeboard.

We have added supplementary instructions on the radar freeboard at the beginning of Sec. 3.1.

[Figure]

Figure 1 Schematic diagram of parameters regarding different freeboards.

**(2) ULS Draft to SIT**

The sea ice draft ( $h_{draft}$ ) from ULS were converted to SIT under hydrostatic equilibrium:

$$h_{si} = \frac{\rho_{sw} h_{draft} - \rho_s h_s}{\rho_{si}}$$

As shown in Figure 11, the ice drafts at the four ULS were larger than 0 at summer time, we therefore assume the ice type as MYI. Consequently, we used a fixed sea ice density of 917 kg/m$^3$ and a seawater density of 1024 kg/m$^3$. The snow depth and density are from W99.

**(3) Adding correlations while comparing**

We have added the mean error (ME) and correlations according to your suggestion in Tables 4, 5, 6, 7.

**(4) Calibration of ERS-2**

We acknowledge the issue of waveform blurring over sea ice for ERS-2. Our data processing methods inherently address this problem to some extent.

Regarding lead detection, we use a combination of waveform parameter thresholds and the lowest elevation method (LEM). For ERS - 2, we use the pulse peakiness (PP) parameter for lead identification. However, we are aware that the waveform blurring may affect the accuracy of this method, especially in thin ice-covered areas. The LEM, which is based on the premise that the surface height of leads is lower than that of nearby sea ice, helps to correct for misidentifications caused by waveform blurring. This combined approach is a form of correction for the waveform-related issues in ERS-2 data.

As for calibration between ERS-2 and Envisat, while the altimeters on ERS-2 and Envisat are similar in some aspects, we did not conduct a separate calibration specifically for ERS-2 and Envisat as we did for Envisat and CryoSat-2. The reason is that the difference in thickness between ERS-2 and Envisat during their common mission period is approximately -0.39 m, which is negligible compared with the difference between CryoSat-2 and Envisat. We applied the monthly correction grid generated from the Envisat-CryoSat-2 comparison to the ERS-2-based thickness for correction, which also helps to account for any systematic differences related to waveform blurring or other factors between ERS-2 and Envisat. If we apply another calibration between ERS-2 and Envisat, the residuals between Envisat and CryoSat-2 will be introduced, which could lead to the superposition of multiple errors.

**(5) Coverage and Resolution issues**

The discrepancy in spatial coverage is caused by multiple factors, such as changes in sea ice extent and data exclusion. The resolution during the Envisat period appears to be higher than that of CryoSat-2. This is mainly because the Envisat results have more noise, while the Cryosat-2 results are smoother. In fact, we uniformly used a grid with a resolution of 5 km.

The choice of 5 km resolution was made to balance the need for high-resolution details and the availability of data. A 5 km resolution allows us to capture more fine - scale features of sea ice thickness compared to coarser resolutions. Although visually the resolution might seem closer to 25 km in some cases, this could be due to data interpolation and the characteristics of sea ice distribution. We believe that the 5 km resolution provides valuable information, especially in areas with complex sea ice dynamics. However, we also understand the advantages of a 12.5 km resolution, such as better coverage and potentially less noise. In future studies, we will consider using a 12.5 km resolution as an alternative and compare the results to further explore the optimal resolution for Arctic sea ice thickness monitoring.

**(6) On the parameters of $a_5$ and $a_6$**

As explained in $SP(20)$, $a_5 = 1 - \dfrac{\rho_{si}}{\rho_{sw}}$ and $a_6 = (1 - \dfrac{\rho_s}{\rho_{sw}} - \theta)h_s$. $a_5$ is a combination of densities of sea ice and seawater, while $a_6$ is much more complex. In Xiao et al. (2020), we derived sea ice density from $a_5$ by setting seawater density as a fixed value (1024 kg/m³). Figure 2 shows the sea ice density distribution for the 2018 -2019

Arctic sea ice growth season from October to April. It can be easily found that the thin ice density is larger than thick ice density. Thin ice density ranges from 915 ~ 920 kg/m³, while the thick ice density ranges from 880 ~ 885 kg/m³.

[Figure]

Figure 2 Arctic sea ice density with the LSA method for the 2018 – 2019 Arctic sea ice growth season from October to April.

**(7) On the projection**

In this study, we used the NSIDC's Polar Stereographic Projection, detailed information can be found in https://nsidc.org/data/user-resources/help-center/guide-nsidcs-polar-stereographic-projection.

In the .nc file, the key parameters of this projection are set as follows: +proj=stere +lat_0=90 +lat_ts=70 +lon_0=-45 +k=1 +x_0=0 +y_0=0 +a=6378273 +b=6356889.449 +units=m +no_defs.

In the manuscript, we used the Generic Mapping Tools (GMT, https://www.generic-mapping-tools.org/) to present the distributions of SIT. The following command was used:

gmt plot -R-180/180/60/88 -Js0/90/2i/45 *thicknessfile* -Sp -C

Specifically, we adopt the polar stereographic projection with a central longitude of 0° and a central latitude of 90°, which means we choose the North Pole as the projection center. The standard parallel of this projection is set at 45°.

**Specific Comments**

1. l.24 : The only mention of the projection used. Should be specify within the paper with all the requested parameters to use this projection, in particular the True Latitude (lat_ts). Also we would recommend to use the EASE v2 https://nsidc.org/data/user-resources/help-center/guide-ease-grids, which becomes the reference and which is much more convenient as the resulting grid is a regular Cartesian grid in meters and allows to compute directly distances, surfaces and volumes in ISU, which is not the case for the stereopolar projection.

**Response:** Thanks for your suggestion. As you mentioned, we indeed use the NSIDC's Polar Stereographic Projection, detailed information can be found in https://nsidc.org/data/user-resources/help-center/guide-nsidcs-polar-stereographic-projection.

In the .nc file, the key parameters of this projection are set as follows: +proj=stere +lat_0=90 +lat_ts=70 +lon_0=-45 +k=1 +x_0=0 +y_0=0 +a=6378273 +b=6356889.449 +units=m +no_defs.

Regarding the EASE v2 projection you recommended, we understand that it has many advantages. For example, the resulting grid is a regular Cartesian grid, which allows for direct calculation of distances, areas, and volumes in ISU, making it more convenient to use. However, our choice of the Polar Stereographic Projection is mainly due to the focus of our research on sea-ice-related applications. The Polar Stereographic Projection has unique advantages in sea-ice research in polar regions. It

is tangent to the Earth's surface at 70°N/S, resulting in minimal grid distortion near the marginal ice zone. This is crucial for accurately analyzing the distribution, movement, and change characteristics of sea ice. Many NSIDC-archived datasets, including a large number of brightness temperature and sea-ice products, also use this projection, which demonstrates its reliability and applicability in the field of sea-ice research. It might be feasible for us to provide an alternative data version in the EASE grid format. This could potentially meet the diverse requirements of users who prefer or require data in this particular grid system for their analysis and research work.

2.  l.46: please specify that the Arctic could become ice free in less than a decade in summer.

**Response:** The statement has been modified.

3.  l.95: I was surprised by these snow depth reduction announced over FYI and MYI but in fact in Webster et al. 2014, they do not speak about FYI/MYI but about specific seas: "For the 2009–2013 period, the products show that snow has decreased by 37 ± 29% in the western Arctic and by 56 ± 33% in the Beaufort and Chukchi seas, compared to the 1954 - 1991 snow depth climatology produced by W99." Please correct it or provide the original citation if any.

**Response:** It has been corrected.

4.  l.148: I suppose there is an error in this sentence: "coarse across-track spacing of 25 km at 75° and 4 km at 60° provided by ERS-2 and Envisat" as the across track spacing decreases with the latitude.

**Response:** This sentence has been corrected as: The across-track spacing of CryoSat-2 is approximately 2.5 km at 75° and 4 km at 60°, which is a significant improvement compared with the coarse across-track spacing of 25 km at 75° and 40 km at 60° provided by ERS-2 and Envisat.

5.  l.152: CryoSat-2 orbit is not any more a repeat cycle since several years, however there is still a sub-cycle of about 30 days.

**Response:** It is revised.

6. l.185,248,270,285,286,293,294: It's very pertinent to recall the uncertainties for each product. I would just recommend to always use the same units (meters or centimetres).

**Response:** The units have been unified.

7. l.195: CryoSat-2 ICE baseline-E L1B is the official ESA product, not an AWI product. Please specify if you use the AWI product or the ESA product.

**Response:** This statement is introducing the AWI-CS2 product. According to the Product User Guide (Hendricks and Paul, 2023), CryoSat-2 ICE baseline-E L1B data are the input data for AWI-CS2 sea ice thickness product.
Hendricks, S. and Paul, S. (2023): Product User Guide & Algorithm Specification - AWI CryoSat-2 Sea Ice Thickness (version 2.6), https://doi.org/10.5281/zenodo.10044554.

8. l.238: This longtime series is described in Bocquetetal. 2024 https://doi.org/10.1029/2023JC020848. Bocquet et al. 2023 explains the methodology and is only over Arctic from 1995 to 2021 (without ERS1).

**Response:** The reference has been updated.

9. l.249: "RMSE of 12-28 cm for Envisat period and 15-21 cm for CryoSat-2 period (Guerreiroetal., 2017)." Guerreiro or Bocquet? It's not the sametime series.

**Response:** This sentence has been revised as:
The draft of CTOH, compared with ULS measurements in BGEP and in Fram Strait, was overestimated by about 0.2 m for CryoSat-2 period and underestimated by 0.11 m and 0.16 m for Envisat and ERS-2, respectively (Bocquet et al., 2024).

10. l.300: DTU18MSS is endowed with discontinuity problems close to the MIZ and you should use DTU21MSS. See: "The DTU21 global mean sea surface and first evaluation" in Earth System Science Data 15(9):4065-4075 10.5194/essd-15-4065-2023

**Response:** Thank you for pointing out this issue. When we initiated this work, DTU21MSS had not been released yet. Given the discontinuity problems of DTU18MSS near the MIZ, we fully recognize the importance of using DTU21MSS. In our next step, we have planned to upgrade our product and will definitely adopt DTU21MSS. This will not only help us address the existing issues related to the MSS data source but also enhance the overall quality and reliability of our product.

11. l.366: The MSS includes the geoid (MSS=geoid+MDT). Also I'm surprise that you don't correct for the usual altimetry corrections such as the wet tropo, the dry tropo, the ocean tide, load tide, pole tide, DAC, etc. You would get even more flat measurements.

**Response:** The geoid undulations and mean dynamic topography (MDT) were removed in the relative elevation. Geophysical corrections were applied using the models or datasets provided in CryoSat-2/Envisat/ERS-2 product before subtracting the MSS height.

12. l.367: I did not know about Pauta criterion, I think it would be interesting to explain it here in few words.

**Response:** The Pauta Criterion, also known as the 3σ rule, is a statistical method used to identify outliers in a data set. It is based on the characteristics of the normal distribution. In a normal distribution, about 99.73% of the data lies within the interval of the mean plus or minus three standard deviations ($\mu \pm 3\sigma$). According to this criterion, data points that fall outside this range ($x < \mu - 3\sigma \ or \ x > \mu + 3\sigma$) are considered outliers.
Reference: Shi, H., Guo, J., Deng, Y. et al. Machine learning-based anomaly detection of groundwater microdynamics: case study of Chengdu, China. Sci Rep 13, 14718 (2023). https://doi.org/10.1038/s41598-023-38447-5.

13. l.375: interpolated is between 2 measures, here I would say extrapolated.

**Response:** Thanks for your advice, it has been revised.

14. l.380-388: this part is not clear at all because you mixed-up the ice-freeboard ($FB_i$, the real freeboard of the ice), the radar-ku-freeboard ($FB_{ku}$, the freeboard

measured by the radar) and the total-freeboard ($FB_t$, ice+snow freeboard that is measured by the lidar of OIB or ICESat-2). They are linked by the following relations: $FB_t = FB_i + SD$ and $FB_{ku} = FB_i - (c_v/c_s - 1) \times SD$, where $SD$ is the Snow Depth and $c_v/c_s$ is the ratio of the speed of light in vaccum and in snow. This ratio depends on the snow density $\rho_{snow}$. From Ulaby 1986 we have: $c_v/c_s = (1 + 0.00051 \times \rho_{snow})^{1.5}$ (Tiuri et al. 1984 suggest: $c_v/c_s = (1 + 1.7\rho_{snow} + 0.7\rho_{snow}^2)^{0.5}$. Each time you speak about freeboard you must specify which freeboard you are speaking about.

**Response:** As shown in the following figure, we define the terminology of freeboard:

- ice freeboard ($FB_i$): refers to the elevation of the snow–ice interface above the local sea level;

- total freeboard ($FB_t$): refers to the elevation of the air–snow interface above the local sea level, which is sensed by laser altimetry;

- radar freeboard: as the radar waves do not fully penetrate snow above ice, we here define the term radar freeboard as the elevation of penetration interface above the local sea level (Ricker et al., 2014).

As for the ice freeboard the lower wave propagation speed in the snow layer requires a correction, the radar-ku-freeboard ($FB_{ku}$) is defined, but is not applied for the radar freeboard in this study. Therefore, the freeboard mentioned in this study refers to radar freeboard.

We have added supplementary instructions on the radar freeboard at the beginning of Sec. 3.1.

[Figure]

Figure 1 Schematic diagram of parameters regarding different freeboards.

15. l.381: "The OIB **total (or lidar)** freeboard was modified with snow depth". Here you should also specify if you have just removed the snow depth to get an **ice freeboard** or if you also have corrected for the speed propagation to get a **radar-ku-freeboard** for the following comparisons.

**Response:** The OIB total freeboard was firstly modified to ice freeboard using snow depth from the snow radar before comparison.

16. l.382: "The mean **radar? ice? yours from OIB? yours from satellite?** freeboard along this track in this study"

**Response:** Here it refers to the mean radar freeboard.

17. l.383: "while the mean **radar** freeboard from the Baseline E".

**Response:** It is revised.

18. l.384: "The mean value of the modified OIB freeboard was 0.261" -> The mean value of the **ice? radar?** freeboard obtained from OIB was 0.261 m.

**Response:** Here it refers to the ice freeboard.

19. l.387: The following sentence is wrong: "the waveform threshold method leads to an underestimation of the freeboard ... ". The threshold method sometimes overestimates the FB and sometimes underestimates it as you show it later on in this paper. It mainly depends on the roughness of the ice, i.e. on ice type (FYI/MYI).

"*... which explains why the freeboard in the Baseline E product was smaller than our estimates and the modified OIB freeboard.*" No, this is explained by the following equation $FB_{ku} = FB_i - (c_v/c_s - 1) \times SD$, which shows that the **radar freeboard is always smaller than the ice freeboard** and it can even be negative for small $FB_{ice}$ and large $SD$.

**Response:** The freeboard in Baseline E product refers to the radar freeboard and is computed as: [radar_freeboard_20_ku] = [height_1_20_ku] - [ssha_interp_20_ku]. A

correction for pulse delay due to snow depth is provided in [snow_depth_cor_20_ku] but is not applied.

Also, as explained above, the radar freeboard in this study refers to the elevation of penetration interface above the local sea level, while the ice freeboard refers to the elevation of the snow–ice interface above the local sea level. Therefore, the radar freeboard will be larger than the ice freeboard.

As shown in Figure 3 in the revision, when the radar burst is reflected from thin ice, specular echoes occur and will be misidentified as leads. Thus, an overestimation will occur on the SSH determination and leading to an underestimation of the ice radar freeboard.

To avoid misconception, this statement has been revised as:

The misidentification of leads in waveform threshold method leads to an underestimation of the radar freeboard, which explains why the freeboard in the Baseline E product was smaller than our radar freeboard.

Reference: CryoSat Ice netCDF L2 Product Format Specification, Issue 2.1. IPF1 L1B Product Formats (esa.int)

20. l.406: This equation is not the equation of the hydrostatic equilibrium between FBi, SIT and SD!

    Here it is: $FB_i = SIT(1 - \rho_{ice}/\rho_{water}) - SD \times \rho_{snow}/\rho_{water}$

    What you have written is the equation that links $FB_{ku}$ with $SIT$ and $SD$ (ie, what you call $h_{fb}$ is in fact $FB_{ku}$ and what you call theta is $c_v/c_s$ including possibly a penetration factor $P * c_v/c_s$).

**Response:** Under the assumption of hydrostatic equilibrium, sea ice thickness can be calculated as (Ricker et al., 2014; Tilling et al., 2018):

$$h_{si} = \frac{\rho_{sw}}{\rho_{sw} - \rho_{si}} h_{fb\_ice} + \frac{\rho_s}{\rho_{sw} - \rho_{si}} h_s \tag{1}$$

where $h_{si}$ is the sea ice thickness; $h_s$ is the snow depth on sea ice; and $\rho_{sw}$, $\rho_{si}$, and $\rho_s$ are the densities of sea water, sea ice, and snow, respectively. We have to be aware that $h_{fb\_ice}$ here refers to the ice freeboard.

As the radar signal cannot penetrate the snow thoroughly, we defined the radar freeboard in this study. As shown in Figure 1 and Figure 2 below, the radar freeboard

( $h_{fb}$ ) refers to the elevation of penetration interface above the local sea level, and $h_{ps}$ is the penetration depth of radar signals. The model for the conversion of freeboard to thickness can be modified as:

$$h_{si} = \frac{\rho_{sw}}{\rho_{sw} - \rho_{si}} h_{fb} + \frac{\rho_s - \rho_{sw}}{\rho_{sw} - \rho_{si}} h_s + \frac{\rho_{sw}}{\rho_{sw} - \rho_{si}} h_{ps}$$
$$= \frac{\rho_{sw}}{\rho_{sw} - \rho_{si}} h_{fb} + \frac{\rho_s - \rho_{sw}}{\rho_{sw} - \rho_{si}} h_s + \frac{\rho_{sw}}{\rho_{sw} - \rho_{si}} \theta \cdot h_s$$

(2)

where $\theta$ is the penetration factor of radar signals.

We firstly model the freeboard as a quadratic function of the local ice surface terrain within the grid:

$$h_{fb}(x, y) = \overline{h_{fb}} + a_0 x + a_1 y + a_2 x^2 + a_3 y^2 + a_4 xy$$

(3)

where $\overline{h_{fb}}$ indicates the mean freeboard of the grid cell and $x$ and $y$ represent the longitudinal and latitudinal surface distances between the observation and the central point of the grid cell, respectively. According to Equation (2),

$$\overline{h_{fb}} = (1 - \frac{\rho_{si}}{\rho_{sw}})\overline{h_{si}} + (1 - \frac{\rho_s}{\rho_{sw}} - \theta)h_s$$

(4)

Thus, Equation (3) can be rewritten as follows:

$$h_{fb}(x, y) = (1 - \frac{\rho_{si}}{\rho_{sw}})\overline{h_{si}} + (1 - \frac{\rho_s}{\rho_{sw}} - \theta)h_s + a_0 x + a_1 y + a_2 x^2 + a_3 y^2 + a_4 xy$$
$$= a_0 x + a_1 y + a_2 x^2 + a_3 y^2 + a_4 xy + a_5 \overline{h_{si}} + a_6$$

(5)

Details of the LSA method are introduced in Xiao et al. (2020).

[Figure]

Figure 2 Schematic of the radar altimeter observing the sea ice thickness

21. l.407-409: Please specify which are the inputs and which are the unknown. It looks like that $h_{fb}$, x and y are the inputs and a0-a7 and $h_{si}$ are the 8 unknown, right?

**Response:** Yes, $h_{fb}$, $x$ and $y$ are the inputs, $a_0 - a_6$ and $\overline{h_{si}}$ are the 8 unknown parameters.

22. l.410: x and y are really lat and lon? It's strange with your grid projection. Using EASE2 they could be directly in meters ;-)

**Response:** $x$ and $y$ represent the longitudinal and latitudinal surface distances between the observation point and the central point of the grid cell.

23. l.418: I suppose it's 25km here, not 10km. as you have just computed the 25km x 25km grids. However, when you look at the data supplied, the maps appear very patchy. Can you explain this? (not enough measurements, even at 25km resolution?).

**Response:** Thank you for bringing this to our attention. It should be 25 km as we've been working with 25 km x 25 km grids.
Regarding the patchy appearance of the maps, it is indeed due to insufficient measurements. Despite the 25 km resolution, the data collection in certain areas has been limited, as we need at least 8 observations to figure out the SIT in a certain grid. Although, we can use interpolation methods to fill these gaps, but it will bring new error sources. In the future study, we will reduce the number of necessary observations by fixing some parameters (such as the seawater density), thereby decreasing the amount of blank data.

24. l.426: typo: antimeres -> altimeters

**Response:** It has been corrected.

25. l.427: "The pulse-limited altimeters have a large footprint of 2–10 km" radius or diameter?

**Response:** The pulse-limited altimeters have a large footprint of 2–10 km in diameter.

26. l.441: typo: Bocquest -> Bocquet

**Response:** It has been corrected.

27. l.442: "for calibrating freeboard measurements from Envisat and ERS-2."   -> for calibrating Envisat freeboard measurements from CryoSat-2 and ERS-2 from calibrated Envisat.

**Response:** It has been corrected.

28. l.443: "Tilling et al. (2019) developed a physical-based approach to correct Envisat SIT … " Could be confusing with the retracker physical-based approach and it's not more physical than considering the ice roughness as it is usually done, I would avoid this term.

**Response:** Thanks for your advice. This statement has been revised as:
Tilling et al. (2019) developed a physical-based approach to corrected Envisat SIT according to the relationship between the thickness differences between Envisat and CryoSat-2 and the along-track distance between leads and the closest floe in the Envisat measurements.

29. l.453: I do not agree with this conclusion: "Compared with CryoSat-2 thickness, Envisat thickness showed an overestimation of $0.19 \pm 0.67$ m in January 2011." As it is shown in your maps and histograms, LRM gets thinner ice over thin ice and thicker over thick ice relatively to SAR as it was explained in Laforge et al 2021 https://doi.org/10.1016/j.asr.2020.02.001

**Response:** Thanks for your comments. Indeed, the original expression was not quite accurate. The overestimation refers to the overall difference between CS-2 and Envisat thickness.

30. l.473, Table 5: Is 'Mean' the Mean Bias? Is STD the STD of the difference or of the product? Would be very pertinent to add the Correlations.

**Response:** Yes, Table 5 is the statistics of the mean values and STDs of the difference between Envisat and CryoSat-2 thickness during the common mission period. We have added the correlations in the Table.

31. l.480: The main problem with ERS-1 and ERS-2 is related to the blurring of their waveforms over sea ice. You don't mention it and it looks like that you don't have applied specific correction for this problem. Any other calibration between ERS2 and Envisat, as for Envisat versus CryoSat2? As it is an important problem it would also be important to see the maps you obtain in front of Envisat map for the same period. And once again the correlation is an important criteria that should also be added.

Response: Thanks for your valuable comments and suggestion. We acknowledge the issue of waveform blurring over sea ice for ERS-2. Our data processing methods inherently address this problem to some extent.

Regarding lead detection, we use a combination of waveform parameter thresholds and the lowest elevation method (LEM). For ERS - 2, we use the pulse peakiness (PP) parameter for lead identification. However, we are aware that the waveform blurring may affect the accuracy of this method, especially in thin ice-covered areas. The LEM, which is based on the premise that the surface height of leads is lower than that of nearby sea ice, helps to correct for misidentifications caused by waveform blurring. This combined approach is a form of correction for the waveform-related issues in ERS-2 data.

As for calibration between ERS-2 and Envisat, while the altimeters on ERS-2 and Envisat are similar in some aspects, we did not conduct a separate calibration specifically for ERS-2 and Envisat as we did for Envisat and CryoSat-2. The reason is that the difference in thickness between ERS-2 and Envisat during their common mission period is approximately -0.39 m, which is negligible compared with the difference between CryoSat-2 and Envisat. We applied the monthly correction grid generated from the Envisat-CryoSat-2 comparison to the ERS-2-based thickness for correction, which also helps to account for any systematic differences related to waveform blurring or other factors between ERS-2 and Envisat. If we apply another calibration between ERS-2 and Envisat, the residuals between Envisat and CryoSat-2 will be introduced, which could lead to the superposition of multiple errors.

We have added the correlations in Table 4 according to your suggestion.

32. l.484: typo: gird -> grid

**Response:** It is corrected.

33. l.487, caption Fig 8: what is kermesinus?

**Response:** It is revised to red.

34. l.497: "The sea ice extent did not show any significant changes during this growth." How do you determine the sea ice extent? It's from NSIDC-0051 with concentration>75%? Would worth to recall it here. It is a very important point as the mean SIT highly depends on it (if you consider or not the thin ice in MIZ).

**Response:** As introduced in Sec. 2.4, we define ice floe regions as those with a sea ice concentration in NSIDC-0051 greater than 75%. We have recalled it here.

35. l.505: what do you mean by "normal distribution"?

**Response:** A normal distribution, also known as a Gaussian distribution, is a probability distribution that is symmetric about the mean. In a normal distribution, the data is distributed in a bell - shaped curve.

36. l.510: what do you mean by "sinistrality"?

**Response:** It is revised to left – skewed.

37. l.535: "The mean MYI thickness decreased by 0.017 m/yr during the research period,". Please provide explicitly the period.

**Response:** The research period refers to the period from 1995/1996 to 2022/2023.

38. l.538-570: I'm not fully convinced of the interest of mean SIT for all the Arctic as this value mainly depends on the considered sea ice extent. For instance it can remains about no ice but only some remaining fast ice at the coast to obtain large mean SIT but it means nothing. To make this type of comparison meaningful you could for instance always consider the same mask (region) for each given month.

Or an alternative would be to compute to total volume instead of the mean SIT. However I will not ask you to change this, but at least you should explain specifically how you define the mask, for instance do you always use the area provided by NSIDC-0051 with concentration>75% for each month of each year?

**Response:** Thanks for your suggestion. I do completely agree your opinions on applying the same mask. In the current version, the mask was defined by NSIDC-0051 with concentration > 75% for each month of each year. We will have a further analysis on the Arctic sea ice variation by presenting the total volume in our next step research, as this study is focused on presenting a new Arctic SIT product.

39. l.574: Please specify if you use the same mask for all the products. It's important to make them comparable.

    l.582: The information of the mask is even more important for the CS2SMOS product because it can cover larger region as it also considers thin ice at the margins thanks to SMOS.

**Response:** When comparing different products, we exclusively calculated the differences for those grids in which every product had values. For grids where any one of the products has a missing value, the difference within that grid cell was not computed.

40. l.592, Table 6: The mean bias and the correlation should be added.

**Response:** Thanks for your suggestion. We have added the statistics of ME and R for the comparison. Our product has the highest correlation with CPOM, reaching 0.937. Its correlation with the products of AWI-CS2, GSFC-CS2 and AWI-CS2+SMOS also exceeds 0.9, while the correlation with CCI is the lowest.

41. l.598: it's really important to know which ice density and which snow depth you have chosen to convert the draft to SIT. If these values are not coherent with the product you compare you will necessarily get higher differences for this product. Please also provide the used equation and the input parameters (mainly ice density, SD is based on WC or MWC?).

**Response:** The sea ice draft ( $h_{draft}$ ) from ULS were converted to SIT under

hydrostatic equilibrium:

$$h_{si} = \frac{\rho_{sw} h_{draft} - \rho_s h_s}{\rho_{si}}$$

As shown in Figure 11, the ice drafts at the four ULS were larger than 0 at summer time, we therefore assume the ice type as MYI. Consequently, we used a fixed sea ice density of 917 kg/m$^3$ and a seawater density of 1024 kg/m$^3$. The snow depth and density are from W99.

42. l.606: typo: "The STDs of WHU were close …" -> "The STDs of WHU are close …"

**Response:** It is revised.

43. l.618-620, Tables 7 & 8: Please add the Mean Bias and the Correlation in these tables. Indeed, if there is a significant bias, both the MAE and the STD will be high, but if the correlation is good it will indicate that the tendencies are coherent, which is the most important point to study change rate. (Also it is not necessary to recall the units in each column ;-).

**Response:** We have added the Mean Bias and the Correlations. The correlations between ULS thickness and the products prior to October 2010 are substantially lower compared to those after October 2010. This disparity can be attributed to the limited time series available for the pre-October 2010 period. The correlations of WHU after October 2010 at the three ULSs all exceed 0.7, demonstrating a significant correspondence between the two datasets.

44. l.624: "Then, the mean thickness of the OIB within the grid was compared with the corresponding grid values." OIB products do not include SIT, how do you compute it? Please provide the equation and the input parameters used from OIB data.

**Response:** The IceBridge L4 and Quick Look Sea Ice Freeboard, Snow Depth, and Thickness products includes the SIT.

45. l.636, Figure 16: For some products the count reaches nearly 1500 and for others it is lower than 300. The shape of the histograms being similar, it means that the number of measurements from one product to another can differ by a factor 3. How can you explain it? Is it because of the resolution of the original product?

**Response:** Yes, this disparity is mainly caused by the resolution of the original product. For example, WHU and CPOM feature a higher resolution of 5 km, the numbers of the two products are larger than those of other products.

46. l.643, Table 9: Please add the correlations.

**Response:** The correlations are added. The correlations between OIB thickness and satellite-based products all exceed than 0.85. This indicates a remarkably strong relationship, suggesting that the OIB thickness values show a high degree of correspondence with the estimations provided by the satellite-based products.

47. l.647: "… error propagation of the input uncertainties including radar freeboard, ice density, snow depth … "

**Response:** It is revised.

48. l.655: "Thus, the uncertainties of the SIT can be calculated by the difference of $h_{si}$ in the last two iterations.". I don't understand why the last two iterations are more relevant than the previous ones. To me, this is more a reflection of the speed of convergence of the LSA than the uncertainties. Please justify this solution. For example, you could more naturally assess the distance between the model and the measurements by calculating the STD or MAE between the model and the measurements.

**Response:** The reason we use the difference of $h_{si}$ in the last two iterations to calculate the uncertainties of the SIT is related to the convergence behavior of the iterative process.
As the iterations progress, the calculated values of the SIT gradually converge towards a stable solution. In the early iterations, the values can fluctuate significantly as the model is still adjusting to find the optimal fit. However, as convergence is approached, the changes between consecutive iterations become smaller. The

difference between the values in the last two iterations thus represents the residual change just before the model reaches its final, or near - final, state.

This residual change can be considered as an indication of the uncertainty in the calculated SIT. It shows how much the value is still changing as the model converges, and this remaining variability is a good measure of the uncertainty associated with the calculated SIT.

While metrics like STD or MAE between the model and the measurements can also provide valuable information about the model-measurement distance, in our case, there is a complication. The data we input into our model is the freeboard, while the outputs are the SITs. Therefore, we are unable to directly calculate the STD and MAE.

49. l.678: Please replace the link https://www.legos.omp.eu/ctoh/fr/produits-ctoh/ by a more direct one: http://dx.doi.org/10.6096/ctoh_sit_2023_01

**Response:** It is revised.

---

## Author Comment (AC3)

Thank you for your comprehensive and constructive feedback on our manuscript. We have carefully addressed each of your comments to improve the clarity, accuracy, and overall quality of our work. Below is a summary of the key revisions made in response to your suggestions:

1. **Clarification of Dataset Improvements**: We elaborated on how our Arctic Sea Ice Thickness (SIT) product advances upon previous datasets, highlighting its higher spatial resolution (5 km grid), extended temporal coverage (1995–2023), and innovative data processing methods, including improved lead detection and inter-mission bias correction.

2. **Figure Revisions**: We updated several figures to improve clarity, including reorganizing subfigures, adding color-coded borders, increasing font sizes, and adjusting axis labels. High-resolution versions of all figures were provided in the WORD file.

3. **Validation Section Placement**: We moved the validation section earlier in the manuscript to ensure a logical flow, allowing readers to understand the validation of our methods before delving into the analysis.

4. **Statistical Evaluation Enhancement**: We added Mean Error (ME) and Correlation Coefficient (R) to Table 5 to provide a more comprehensive assessment of our product's performance, demonstrating strong correlations with other datasets.

We sincerely appreciate your thoughtful and detailed feedback, which has significantly strengthened our manuscript. We hope these revisions address your concerns and enhance the clarity, rigor, and impact of our study. Thank you for your invaluable contributions to improving our work.

Next, we respond point by point to your comments.

**General comments:**

1. It is currently difficult to identify how this dataset improves upon previous SIT datasets. The authors need to articulate these improvements more clearly.

**Response:** Thanks for your suggestion. In this study, we have developed a new Arctic

SIT datasets by combining multiple radar altimetry data from ERS-2, Envisat, and CryoSat-2. The SIT is presented on a monthly 5 km grid, which is highest in satellite altimetry-based SIT products. The time series of our product spans from 1995 to 2023, making it the second-longest continuous record, surpassed only by the CTOH dataset. Furthermore, we have introduced an innovative data processing methodology that encompasses lead detection, freeboard-to-thickness conversion, and inter-mission bias correction. Specifically, we enhanced the lead detection method by combining waveform parameter thresholds with the lowest elevation approach. This improved method effectively mitigates the impact of grease ice, nilas, and newly frozen leads. The freeboard was then converted to thickness using a quadratic model based on hydrostatic equilibrium and least squares adjustment. Additionally, we generated a monthly thickness correction grid using common period observations from Envisat and CryoSat-2 to address inter-mission biases. This correction reduced the thickness difference between Envisat and CryoSat-2 from 0.66 m to 0.35 m.

We have detailed these advancements in the conclusion section (Lines 768-781) to provide a clearer understanding of the improvements our dataset offers over previous SIT datasets.

2. The language and structure of the paper require refinement to enhance overall readability.

**Response:** We have engaged a professional scientific editor who is proficient in English to review the language of the paper. The editor has checked for grammar errors, improved sentence structures, and ensured the clarity of the writing. Additionally, we have re-organized the paper to enhance its logical flow.

3. The figures require better visual design to improve clarity and aesthetics.

**Response:** Thank you for your constructive feedback on the figures. We have carefully revised and polished the figures based on your specific comments to enhance their clarity and visual appeal. Additionally, we have provided high-resolution versions of the figures in the WORD file to ensure better readability

and detail. We hope these improvements meet your expectations and contribute to a more effective presentation of our data.

**Specific comments:**

4. Line 22: "Finally, the monthly SIT estimates for the Arctic Ocean from October 1995 to December 2023 are generated." This statement is unclear. You only obtained SIT results from October to April of the following year. I understand that it is challenging to extract sea ice thickness during the summer, but this statement is misleading.

**Response:** Thank you for pointing this out. We have revised the statement in Lines 23-24 to clarify the temporal scope of our SIT estimates. The updated text now reads: "Finally, the monthly SIT estimates for the Arctic Ocean are derived for the freezing period spanning from October 1995 to December 2023."

5. Lines 85-100: What is the purpose of describing the progress in snow depth research?

**Response:** Thanks for your question. The discussion of snow depth research progress serves a critical purpose in our study. Snow depth is a key factor that significantly impacts the accuracy of SIT estimates, as uncertainties in snow depth can contribute up to 70% of the total uncertainty in SIT calculations. In previous studies, snow depth data from sources such as W99 or PMW sensors were commonly used. However, these datasets are known to have substantial uncertainties in snow depth estimation.

By highlighting these limitations, we aim to provide context for our methodological choice. Specifically, we introduced the Least Squares Adjustment (LSA) method to convert freeboard to thickness, which addresses the inherent uncertainties in traditional snow depth datasets. This approach allows us to improve the reliability and accuracy of our SIT estimates. We hope this clarification underscores the importance of our methodological innovation in the broader context of snow depth research.

6. Line 111: "They have limited temporal coverage", what is the temporal coverage?

**Response:** Thank you for your question. To provide clarity, we have added specific details about the temporal coverage of ICESat and ICESat-2 in Lines 112-113. The revised text now reads: "They have limited temporal coverage, with ICESat operating from 2003 to 2009 and ICESat-2 commencing operations in 2018."

7. The authors produced a SIT product with a temporal resolution of one month and a spatial resolution of 5 km. Lines 101-116: The authors should explain the issues with current products in terms of spatiotemporal resolution and coverage.

**Response:** Thanks for your valuable suggestion. The temporal coverage of SIT product is inherently linked to the operational periods of the satellites used. Most existing SIT products primarily provide data during the CryoSat-2 era, as earlier satellites like ERS-2 and Envisat are pulse-limited and have larger footprints. These larger footprints are more susceptible to specular returns, leading to increased mixing of different surface types. This susceptibility poses significant challenges for lead identification and freeboard retrieval, ultimately affecting the spatial resolution of SIT products.

In this study, we addressed these challenges by enhancing the lead detection method through the combined use of waveform parameter thresholds and the lowest elevation approach. This improved method effectively mitigates the impact of grease ice, nilas, and newly frozen leads, thereby improving the accuracy and reliability of our SIT product.

We have added detailed explanations regarding these issues in Lines 115-123 to provide a clearer understanding of the limitations of current products and the advancements made in our study. We hope this addition enhances the clarity and depth of our manuscript.

8. Line 138: A reference is required here.

**Response:** We have added the following reference in Lines 143:

Legresy, B., Papa, F., Remy, F., Vinay, G., Van Den Bosch, M., and Zanife, O. Z.: ENVISAT radar altimeter measurements over continental surfaces and ice caps using

the ICE-2 retracking algorithm, Remote Sens Environ, 95, 150–163, https://doi.org/10.1016/J.RSE.2004.11.018, 2005.

9. There is no textual reference to Table 1 in the manuscript. It should be moved to the appendix.

**Response:** Thanks for your suggestion, Table 1 has been moved to the appendix.

10. Table 2 should be moved to Line 164.

**Response:** Thank you for your suggestion. We have relocated Table 2 (now referred to as Table 1 in the revised manuscript) to the recommended position at Line 172.

11. Figure 2 is unclear: Figures (c)-(g) should be placed below (b) according to the sequence; The connections between the boxes in (a) and (b) and figures (c)-(g) are unclear. I suggest adding color-coded borders for each subfigure, corresponding to the colored boxes and arrows in (a) and (b); The font size of the coordinates in (a) and (b) is too small, and the y-axis label in (b) is reversed.

**Response:** Thank you for your detailed feedback on Figure 2 (now referred to as Figure 3 in the revised manuscript). We have made the following updates to address your concerns:

- Reorganization: Figures (c)-(g) have been repositioned below (b) to follow a more logical sequence.

- Clarification of Connections: We have added color-coded borders to each subfigure, corresponding to the colored boxes and arrows in (a) and (b), to make the connections clearer.

- Font Size and Axis Labels: The font size of the coordinates in (a) and (b) has been increased for better readability. Additionally, to prevent the y-axis labels from overlapping, the y-axis of (b) has been moved to the right-hand side.

We hope these improvements enhance the clarity and overall presentation of the figure.

12. Figure 4 appears to be a screenshot rather than an original figure. The titles for figures (a)-(c) seem incomplete.

**Response:** Thank you for pointing this out. We acknowledge that the resolution of Figure 4 in the PDF version was compromised, which may have given the impression of a screenshot. To address this, we have provided high-resolution versions of the figure in the WORD file to ensure clarity and detail.

Additionally, we have updated the caption for Figure 4 (now referred to as Figure 5 in the revised manuscript) to include more complete explanations for subfigures (a)-(c). The revised caption now reads: " Numbers of (a) ERS-2, (b) Envisat, and (c) CryoSat-2 observations falling within each 5 km grid cell.

13. I understand that the primary focus of this paper is to publish a new SIT product. However, I hope you can provide explanations for sea ice variation phenomena in the results section. For instance, in Line 530, why was the average SIT in 2012/2013 the historical minimum? This could be explained by citing relevant literature.

**Response:** Thank you for your insightful suggestion. We have incorporated additional explanations in Lines 659-664 to address the sea ice variation phenomena observed in our results. Specifically, we have added the following text:

Correspondingly, the Arctic sea ice cover reached a record minimum in 2012 for the satellite era. Cui et al. (2015) demonstrated that in 2007 and 2012, there was a higher surface air temperature and sea level pressure, which was accompanied by increased surface specific humidity and a higher sea surface temperature. As a result, the strengthened poleward wind was conducive to the melting of summer Arctic sea ice in various regions during those two years.

This addition provides a clearer understanding of the factors contributing to the historical minimum in average SIT during 2012/2013, supported by relevant literature. We hope this enhancement adds depth to our discussion and better contextualizes the observed phenomena.

14. Figures 11-15: All these figures are line plots of SIT. Why do some include grids while others do not? Additionally, the x-axis title is "Year" in all cases, but some display "22/23," while others use "2023." My suggestion is to either unify the format or explain the differences.

**Response:** Thank you for pointing this out. To address the inconsistencies in Figures 11-15 (now referred to as Figures 10, 16, 17, and 18 in the revised manuscript), we have made the following clarifications:

· Grids: Grids are included in Figure 18 (previously Figure 13) to enhance readability.

· X-axis Format: As defined in Lines 515-516, the annual average SIT refers to the average thickness during the frozen season, specifically from October to April of the following year. In Figures 10, 16, and 17 of the revision, we present the variations of annual average SIT, hence the x-axis is displayed as "22/23" to indicate the frozen season spanning two calendar years. In contrast, Figure 18 shows the monthly average thickness from 1995 to 2023, so the x-axis refers to specific years.

We hope this explanation clarifies the differences in the figures and ensures a more consistent and understandable presentation. Thank you for your valuable feedback, which has helped improve the clarity of our manuscript.

15. Logically, validation should precede sea ice thickness analysis.

**Response:** Thank you for your suggestion. We agree that validation should logically precede the analysis of sea ice thickness. Accordingly, we have moved the validation section forward in the revised manuscript. This adjustment ensures a more coherent and logical flow, allowing readers to first understand the validation of our methods and data before delving into the analysis of sea ice thickness.

16. Figure 14: The primary focus is the WHU dataset (red line), but it is currently unclear. I suggest: (1) Increasing the color contrast for the red line. (2) Using dashed lines for other datasets to make WHU stand out.

**Response:** Thank you for your valuable feedback. We have updated Figure 14 (now referred to as Figure 10 in the revised manuscript) to enhance the clarity and focus on the WHU dataset. The following changes have been made:

· Color Contrast: We have increased the color contrast for the red line representing the WHU dataset to make it more prominent.

· Line Style: We have used dashed lines for the other datasets to further distinguish the WHU dataset and ensure it stands out clearly.

We hope these adjustments improve the readability and visual impact of the figure, making it easier for readers to focus on the primary dataset of interest.

17. I cannot understand the statistical evaluation in Table 6: (1) MAE measures the average magnitude of absolute errors. (2) STD measures the variability or dispersion of the data, but what does it aim to express here? (3) ME (Mean Error) is crucial for assessing the direction of errors. (4) R (correlation coefficient) is also important for assessing the linear relationship. Therefore, both ME and R should be added to provide a more comprehensive assessment.

**Response:** Thank you for your detailed feedback on the statistical evaluation in Table 6 (now referred to as Table 5 in the revised manuscript). To provide a more comprehensive assessment, we have made the following updates:

· Mean Error (ME): We have added ME to assess the direction of errors, which is crucial for understanding any systematic bias in our product.

· Correlation Coefficient (R): We have also included R to evaluate the linear relationship between our product and other datasets.

The updated statistics in Table 5 (Line 535) now include MAE, STD, ME, and R. Our product demonstrates a strong correlation with other datasets, with the highest correlation of 0.977 with AWI-CS2 and the smallest correlation of 0.879 with GSFC-IS2. These additions provide a more robust and comprehensive evaluation of our product's performance.

We appreciate your suggestions, which have significantly enhanced the statistical analysis and overall clarity of our manuscript.

18. Line 601: Why is October 2010 used as the dividing line for comparing two periods? Please provide justification for this choice.

**Response:** Thank you for your question. The choice of October 2010 as the dividing line for comparing two periods is based on the transition in the type of altimetry data used for SIT calculations. Before October 2010, SIT was derived from pulse-limited altimetry data from ERS-2 and Envisat. These pulse-limited altimeters have larger footprints and lower accuracy compared to CryoSat-2, which began operations in October 2010.

In the revised manuscript, we have provided a more detailed justification for this choice. We first presented the statistics of the draft difference between Upward-Looking Sonar (ULS) observations and satellite-based products for the entire period from 2008 to 2022. Our analysis indicates that products incorporating Envisat data (CCI, CTOH, and WHU) prior to October 2010 exhibit relatively lower accuracy compared to CryoSat-2-based solutions. This distinction is quantitatively substantiated in Table 7, which presents post-October 2010 statistics showing marked accuracy improvements for these three products when transitioning to CryoSat-2 data. The comparative results clearly demonstrate the enhanced precision of CryoSat-2-derived thickness estimates over Envisat-based methodologies.

These discussions have been added in Lines 553-561 to provide a clearer rationale for the choice of October 2010 as the dividing line and to highlight the improvements in accuracy with the use of CryoSat-2 data. We hope this explanation addresses your concern and enhances the clarity of our manuscript.

19. Figure 15: The x-axis labels are unclear. It is unnecessary to label every tick; you can increase the spacing between tick marks. The most important consideration is to clearly convey the information.

**Response:** Thank you for your feedback. We have updated Figure 15 (now referred to as Figure 11 in the revised manuscript) to improve the clarity of the x-axis labels. We have increased the spacing between tick marks to reduce clutter and improve

readability.

20. I understand your intention with A-D in Table 7 and Table 8, but you need to explain this explicitly in the table caption to ensure clarity.

**Response:** Thank you for your suggestion. We have updated Table 7 and Table 8 (now referred to as Table 6 and Table 7 in the revised manuscript) to ensure clarity. The following changes have been made:

- Table Captions: We have explicitly explained the significance of labels A-D in the captions of both tables. This addition provides a clear understanding of the categories and their relevance to the data presented.

- Table Formatting: We have ensured that the tables are formatted consistently and clearly to enhance readability.

---

## Author Comment (AC4)

We sincerely thank the Reviewer for the constructive comments on the manuscript. The reviewer's insights and suggestions have been extremely helpful in improving the quality of our work. We have thoroughly considered each comment and have made the following responses and revisions. The amendments are marked in red in the revision.

Next, we respond point by point to the comments.

**General Comments**

This paper presents a new time series of Arctic sea ice thickness over nearly 30 years based on altimetry observations with an unprecedented resolution of 5km. This is an important subject because only one other series of this length exists to date, and it may help to confirm the initial results and the relevance of this type of monitoring of the state of the sea ice. The comparison is even more interesting as the method used is new and totally independent of the previous ones. It relies on the multiplication of altimetric freeboard measurements in each 5km x 5km grid cell, to determine the thickness of the ice without preconceived ideas about parameters such as density or snow depth. For these reasons, I believe that this work deserves to be published.

Nevertheless, some important concepts have been overlooked, leading to dubious interpretations of certain results. In particular, the authors do not distinguish between the freeboard measured by Ku-band radar altimeters and the freeboard of the ice, which leads to erroneous equations that mainly affect the validation part. Because snow reduces the speed of propagation of the radar wave, the radar freeboard is necessarily smaller than the ice freeboard, and these 2 freeboards cannot be compared directly. The comparisons with OIB freeboard Figure 3 are therefore questionable. For the comparisons with ULS the used ice density is not specified. No correlation is calculated when comparing with other solutions or within-situ measurements.

While ERS2 is a hard point because of the problem of blurring of its waveforms, no resulting map is shown. Although the data supplied with the paper seems to

give consistent results, it would be useful to present comparisons with Envisat over a common month and to discuss any problems encountered, such as the extent of filtering used (Pauta criterion). Also, very strangely, the quality of the results deteriorates sharply, with extremely moth-eaten maps for the Envisat period from October 2002, even though the Envisat measurements are of much better quality than those of ERS2 and their orbits are the same. Coverage became good again with CryoSat-2 from November 2010, but visually the resolution seemed closer to 25km than 5km. This seems to indicate that you forgot to apply the coarse resolution to Envisat, but also that the 5km resolution is far too low to cover the Arctic basin.

All this raises questions about the choice of such a small resolution (5km), compensated by a relatively coarse resolution (25km). For a future study, wouldn't a resolution of 12.5km be more appropriate? Or do you think that 5km really adds value (relevant signal at this scale)?

Finally I would recommend analysing and exporting in the data the parameters a5 and a6, which depend on the densities and the snow depth. This should allow evaluating the consistency of these parameters, and thus the model used.

From a practical viewpoint I would recommend to use the Lambert-Azimuthal-Equal-Area projection with lon0=0, which becomes the reference in our domain, but it's not mandatory as it represents a lot of work without changing the results (see EASE v2 in https://nsidc.org/data/user-resources/help-center/guide-ease-grids). Strangely, while the figures in the paper show maps with lon0=0, the data are centred on lon0=-45.

**Response:** We appreciate your recognition of the significance of our work and the

potential contributions to the study of Arctic sea ice thickness. Below, we address your concerns in detail:

**(1) Concepts on freeboard**

As illustrated in the following figure, we define the terminology of freeboard:

- ice freeboard ($FB_i$): refers to the elevation of the snow–ice interface above the local sea level;

- total freeboard ($FB_t$): refers to the elevation of the air–snow interface above the local sea level, which is sensed by laser altimetry;

- radar freeboard: since the radar waves do not fully penetrate snow above ice, we here define the term radar freeboard as the elevation of penetration interface above the local sea level (Ricker et al., 2014).

For the ice freeboard, the lower wave propagation speed in the snow layer requires a correction, the radar-ku-freeboard ($FB_{ku}$) is defined, but is not applied for the radar freeboard in this study. Therefore, the freeboard mentioned in this study specifically refers to radar freeboard.

We have added supplementary explanations on radar freeboard at the beginning of Section 3.1 (Lines 335–344) to clarify these definitions.

[Figure]

Figure R1 Schematic diagram of parameters regarding different freeboards.

**(2) Validation with ULS draft**

To avoid potential uncertainties associated with the W99 model in draft-to-thickness conversion, we revised our comparative strategy by directly comparing ULS-measured ice drafts with satellite-derived draft estimates. The draft from satellite-based products is calculated by removing ice freeboard from sea ice thickness

(SIT). Since the AWI-SMOS+CS2 and CPOM datasets do not include ice freeboard parameters, we limited our comparison to the other six products.

**(3) Adding correlations while comparing**

Following your suggestion, we have added mean error (ME) and correlation coefficients to Tables 4-8 to enhance the robustness of our validation.

**(4) Calibration of ERS-2**

We acknowledge the issue of waveform blurring over sea ice for ERS-2. Our data processing methods inherently address this problem to some extent.

Regarding lead detection, we employ a combination of waveform parameter thresholds and the lowest elevation method (LEM). For ERS - 2, we use the pulse peakiness (PP) parameter for lead identification. However, we are aware that the waveform blurring may affect the accuracy of this method, especially in thin ice-covered areas. The LEM, which is based on the premise that the surface height of leads is lower than that of nearby sea ice, helps to correct for misidentifications caused by waveform blurring. This combined approach is a form of correction for the waveform-related issues in ERS-2 data.

Regarding calibration between ERS-2 and Envisat, while the altimeters on these platforms are similar in some aspects, we did not conduct a separate calibration specifically for ERS-2 and Envisat, as we did for Envisat and CryoSat-2. The reason is that the difference in thickness between ERS-2 and Envisat during their common mission period is approximately −0.37 m, which is negligible compared to the difference between CryoSat-2 and Envisat. We applied the monthly correction grid generated from the Envisat-CryoSat-2 comparison to the ERS-2-based thickness, which also helps account for any systematic differences related to waveform blurring or other factors between ERS-2 and Envisat. Introducing an additional calibration between ERS-2 and Envisat could lead to the superposition of multiple errors, particularly residuals between Envisat and CryoSat-2.

We added a discussion section to address the above issues in Lines 704-723.

**(5) Coverage and Resolution issues**

The discrepancy in spatial coverage is caused by multiple factors, such as changes in sea ice extent and data exclusion. The resolution during the Envisat period appears higher than that of CryoSat-2, primarily because the Envisat results contain more noise, while the CryoSat-2 results are smoother. In reality, we uniformly used a grid with a resolution of 5 km.

The choice of 5 km resolution was made to balance the need for high-resolution details and the availability of data. A 5 km resolution allows us to capture more fine - scale features of sea ice thickness compared to coarser resolutions. Although visually the resolution might seem closer to 25 km in some cases, this could be due to data interpolation and the characteristics of sea ice distribution. We believe that the 5 km resolution provides valuable information, especially in areas with complex sea ice dynamics. However, we also recognize the advantages of a 12.5 km resolution, such as better coverage and potentially less noise. In future studies, we will consider using a 12.5 km resolution as an alternative and compare the results to further explore the optimal resolution for Arctic sea ice thickness monitoring.

We added a discussion section to address the above issues in Lines 724-736.

**(6) On the parameters of $a_5$ and $a_6$**

We have already included these two parameters in the updated products.

As explained in $SP(20)$, $a_5 = 1 - \dfrac{\rho_{si}}{\rho_{sw}}$ and $a_6 = (1 - \dfrac{\rho_s}{\rho_{sw}} - \theta)h_s$. $a_5$ is a combination of densities of sea ice and seawater, while $a_6$ is much more complex and depends on additional factor. In Xiao et al. (2020), we derived sea ice density from $a_5$ by setting seawater density as a fixed value (1024 kg/m³). Figure R2 shows the sea ice density distribution for the 2018 -2019 Arctic sea ice growth season from October to April. It can be easily found that the thin ice density is larger than thick ice density. Thin ice density ranges from 915 ~ 920 kg/m³, while the thick ice density ranges from 880 ~ 885 kg/m³.

[Figure]

Figure R2 Arctic sea ice density with the LSA method for the 2018 – 2019 Arctic sea ice growth season from October to April.

**(7) On the projection**

In this study, we used the NSIDC's Polar Stereographic Projection, detailed information can be found in https://nsidc.org/data/user-resources/help-center/guide-nsidcs-polar-stereographic-projection.

In the .nc file, the key parameters of this projection are set as follows: +proj=stere +lat_0=90 +lat_ts=70 +lon_0=-45 +k=1 +x_0=0 +y_0=0 +a=6378273 +b=6356889.449 +units=m +no_defs.

In the manuscript, we used the Generic Mapping Tools (GMT, https://www.generic-mapping-tools.org/) to present the distributions of SIT. The following command was used:

gmt plot -R-180/180/60/88 -Js0/90/2i/45 *thicknessfile* -Sp -C

Specifically, we adopt the polar stereographic projection with a central longitude of 0° and a central latitude of 90°, which means we choose the North Pole as the projection center. The standard parallel of this projection is set at 45°.

**Specific Comments**

1. l.24 : The only mention of the projection used. Should be specify within the paper with all the requested parameters to use this projection, in particular the True Latitude (lat_ts). Also we would recommend to use the EASE v2 https://nsidc.org/data/user-resources/help-center/guide-ease-grids, which becomes the reference and which is much more convenient as the resulting grid is a regular Cartesian grid in meters and allows to compute directly distances, surfaces and volumes in ISU, which is not the case for the stereopolar projection.

**Response:** Thank you for your valuable suggestion regarding the projection used in our study. We appreciate your recommendation to consider the EASE v2 projection, which offers several advantages, particularly in terms of its regular Cartesian grid format that facilitates direct calculations of distances, areas, and volumes in ISU.

In this study, we employed the NSIDC's Polar Stereographic Projection, as detailed in the NSIDC's Polar Stereographic Projection Guide. The specific parameters used in the .nc file are as follows:

+proj=stere +lat_0=90 +lat_ts=70 +lon_0=-45 +k=1 +x_0=0 +y_0=0 +a=6378273 +b=6356889.449 +units=m +no_defs.

The true latitude (lat_ts) is set to 70°, which minimizes distortion near the marginal ice zone, a critical region for sea ice analysis.

While we recognize the benefits of the EASE v2 projection, our choice of the Polar Stereographic Projection was driven by its specific advantages for sea ice research in polar regions. This projection is tangent to the Earth's surface at 70°N/S, ensuring minimal grid distortion in areas of interest, such as the marginal ice zone. This is particularly important for accurately analyzing the distribution, movement, and

changes in sea ice. Additionally, many NSIDC-archived datasets, including brightness temperature and sea ice products, utilize this projection, underscoring its reliability and widespread applicability in sea ice research.

That said, we acknowledge the convenience of the EASE v2 projection for certain applications. To accommodate diverse user needs, we are open to providing an alternative data version in the EASE grid format in future updates. This would allow users who prefer or require data in this grid system to conduct their analyses more effectively.

We appreciate your feedback and will consider incorporating this enhancement to improve the accessibility and utility of our dataset.

2. l.46: please specify that the Arctic could become ice free in less than a decade in summer.

**Response:** Thank you for your suggestion. We have revised the statement on Line 48 to clarify that the Arctic could become ice-free in summer within less than a decade. The updated text now reads:

A recent study revealed that the Arctic could become ice-free in summer in less than a decade even in the lowest-emission scenarios.

3. l.95: I was surprised by these snow depth reduction announced over FYI and MYI but in fact in Webster et al. 2014, they do not speak about FYI/MYI but about specific seas: "For the 2009–2013 period, the products show that snow has decreased by 37 ± 29% in the western Arctic and by 56 ± 33% in the Beaufort and Chukchi seas, compared to the 1954 - 1991 snow depth climatology produced by W99." Please correct it or provide the original citation if any.

**Response:** Thank you for pointing this out. We have carefully reviewed the reference and corrected the statement on Line 96. The revised text now accurately reflects the findings of Webster et al. (2014), which highlight regional snow depth reductions rather than differences between FYI and MYI. The updated sentence reads:

According to Webster et al. (2014), the snow depth during the 2009-2013 period has decreased by $37 \pm 29\%$ in the western Arctic and by $56 \pm 33\%$ in the Beaufort and Chukchi seas, compared to the depth in W99.

4. l.148: I suppose there is an error in this sentence: "coarse across-track spacing of 25 km at 75° and 4 km at 60° provided by ERS-2 and Envisat" as the across track spacing decreases with the latitude.

**Response:** Thank you for catching this error. We have revised the sentence on Line 155 to accurately reflect the across-track spacing of CryoSat-2, ERS-2, and Envisat. The corrected sentence now reads:

The across-track spacing of CryoSat-2 is approximately 2.5 km at 75° and 4 km at 60°, which is a significant improvement compared with the coarse across-track spacing of 25 km at 75° and 40 km at 60° provided by ERS-2 and Envisat.

5. l.152: CryoSat-2 orbit is not any more a repeat cycle since several years, however there is still a sub-cycle of about 30 days.

**Response:** Thank you for pointing this out. We have revised the statement on Line 159 to accurately reflect the current status of CryoSat-2's orbit. The updated sentence now reads:

While CryoSat-2 no longer follows a strict repeat cycle, it still maintains a sub-cycle of approximately 30 days, which enables monthly coverage of Arctic sea ice.

6. l.185,248,270,285,286,293,294: It's very pertinent to recall the uncertainties for each product. I would just recommend to always use the same units (meters or centimetres).

**Response:** Thank you for your suggestion. We have unified the units of uncertainties across the mentioned lines to ensure consistency. All uncertainties are now expressed in meters.

7. l.195: CryoSat-2 ICE baseline-E L1B is the official ESA product, not an AWI product. Please specify if you use the AWI product or the ESA product.

**Response:** Thank you for pointing this out. We have clarified the statement on Line 204 to specify the source of the data used. The revised text now reads:

In the latest version of AWI-CS2 (V2.6), CryoSat-2 ICE baseline-E L1B data serve as the input for the AWI-CS2 sea ice thickness product (Hendricks and Paul, 2023).

Reference:

Hendricks, S. and Paul, S. 2023: Product User Guide & Algorithm Specification - AWI CryoSat-2 Sea Ice Thickness (version 2.6), https://doi.org/10.5281/zenodo.10044554.

8. l.238: This longtime series is described in Bocquetetal. 2024 https://doi.org/10.1029/2023JC020848. Bocquet et al. 2023 explains the methodology and is only over Arctic from 1995 to 2021 (without ERS1).

**Response:** Thank you for pointing this out. We have updated the reference on Line 247 to reflect the correct source.

9. l.249: "RMSE of 12-28 cm for Envisat period and 15-21 cm for CryoSat-2 period (Guerreiroetal., 2017)." Guerreiro or Bocquet? It's not the sametime series.

**Response:** Thank you for catching this inconsistency. We have revised the sentence on Line 256 to clarify the source and provide accurate information. The updated text now reads:

The draft of CTOH, compared with ULS measurements in BGEP and in Fram Strait, was overestimated by about 0.2 m for the CryoSat-2 period and underestimated by 0.11 m and 0.16 m for the Envisat and ERS-2 periods, respectively (Bocquet et al., 2024).

10. l.300: DTU18MSS is endowed with discontinuity problems close to the MIZ and you should use DTU21MSS. See: "The DTU21 global mean sea surface and first evaluation" in Earth System Science Data 15(9):4065-4075 10.5194/essd-15-4065-2023

**Response:** Thank you for your valuable suggestion. We have rigorously re-processed all datasets using the updated DTU21MSS model and systematically regenerated our sea ice thickness product. All affected components of the manuscript—including

figures, tables, and related analyses—have been thoroughly revised and updated to reflect these improvements.

We appreciate your feedback and are confident that this update significantly strengthens the quality of our work.

11. l.366: The MSS includes the geoid (MSS=geoid+MDT). Also I'm surprise that you don't correct for the usual altimetry corrections such as the wet tropo, the dry tropo, the ocean tide, load tide, pole tide, DAC, etc. You would get even more flat measurements.

**Response:** Thank you for your insightful comment. To clarify, the geoid undulations and mean dynamic topography (MDT) were removed in the calculation of relative elevation. Additionally, we applied standard geophysical corrections—including wet tropospheric, dry tropospheric, ocean tide, load tide, pole tide, and dynamic atmospheric correction (DAC)—using the models or datasets provided in the CryoSat-2, Envisat, and ERS-2 products before subtracting the mean sea surface (MSS) height.

These corrections ensure that the measurements are as accurate and flat as possible, minimizing errors and improving the reliability of our results.

12. l.367: I did not know about Pauta criterion, I think it would be interesting to explain it here in few words.

**Response:** Thank you for your suggestion. To provide clarity, we have added a brief explanation of the Pauta Criterion (also known as the $3\sigma$ rule) on Line 382. The revised text now reads:

The Pauta Criterion, also known as the $3\sigma$ rule, is a statistical method used to identify outliers in a dataset. It is based on the characteristics of the normal distribution, where approximately 99.73% of the data lies within the interval of the mean plus or minus three standard deviations ($\mu \pm 3\sigma$). Data points falling outside this range ($x < \mu - 3\sigma$ or $x > \mu + 3\sigma$) are considered outliers (Shi et al., 2023).

Reference:

Shi, H., Guo, J., Deng, Y. et al. Machine learning-based anomaly detection of groundwater microdynamics: case study of Chengdu, China. Scientific Reports 13, 14718 (2023). https://doi.org/10.1038/s41598-023-38447-5.

13. l.375: interpolated is between 2 measures, here I would say extrapolated.

**Response:** Thank you for pointing this out. We have revised the term on Line 395 from "interpolated" to "extrapolated" to accurately describe the process. The updated text now reads:

For sections without identified leads, the local SSH was extrapolated from adjacent sections.

14. l.380-388: this part is not clear at all because you mixed-up the ice-freeboard ($FB_i$, the real freeboard of the ice), the radar-ku-freeboard ($FB_{ku}$, the freeboard measured by the radar) and the total-freeboard ($FB_t$, ice+snow freeboard that is measured by the lidar of OIB or ICESat-2). They are linked by the following relations: $FB_t = FB_i + SD$ and $FB_{ku} = FB_i - (c_v/c_s - 1) \times SD$, where $SD$ is the Snow Depth and $c_v/c_s$ is the ratio of the speed of light in vaccum and in snow. This ratio depends on the snow density $\rho_{snow}$. From Ulaby 1986 we have: $c_v/c_s = (1 + 0.00051 \times \rho_{snow})^{1.5}$ (Tiuri et al. 1984 suggest: $c_v/c_s = (1 + 1.7\rho_{snow} + 0.7\rho_{snow}^2)^{0.5}$. Each time you speak about freeboard you must specify which freeboard you are speaking about.

**Response:** Thank you for your detailed feedback. To clarify the terminology and improve the clarity of this section, we have added supplementary explanations at the beginning of Section 3.1:

• ice freeboard ($FB_i$): refers to the elevation of the snow–ice interface above the local sea level;

• total freeboard ($FB_t$): refers to the elevation of the air–snow interface above the local sea level, which is sensed by laser altimetry;

• radar freeboard: as the radar waves do not fully penetrate snow above ice, we here define the term radar freeboard as the elevation of penetration interface above the local sea level (Ricker et al., 2014).

As for the ice freeboard the lower wave propagation speed in the snow layer requires a correction, the radar-ku-freeboard ($FB_{ku}$) is defined, but is not applied for the radar freeboard in this study. Therefore, the freeboard mentioned in this study refers to radar freeboard.

15. l.381: "The OIB **total (or lidar)** freeboard was modified with snow depth". Here you should also specify if you have just removed the snow depth to get an **ice freeboard** or if you also have corrected for the speed propagation to get a **radar-ku-freeboard** for the following comparisons.

**Response:** Thank you for your clarification. We have revised the statement on Line 400 to specify the process more accurately. The updated text now reads:
The OIB total freeboard was first modified to ice freeboard by removing the snow depth derived from the snow radar before comparison.

16. l.382: "The mean **radar? ice? yours from OIB? yours from satellite?** freeboard along this track in this study"

**Response:** Thank you for pointing out this ambiguity. We have revised the statement on Line 402 to clarify that it refers to the mean radar freeboard. The updated text now reads:
The mean radar freeboard along this track in this study was approximately 0.280 m, while the mean radar freeboard from the Baseline E product was 0.238 m.

17. l.383: "while the mean **radar** freeboard from the Baseline E".

**Response:** Thank you for your feedback. We have revised the statement on Line 403 for clarity.

18. l.384: "The mean value of the modified OIB freeboard was 0.261" -> The mean value of the **ice? radar?** freeboard obtained from OIB was 0.261 m.

**Response:** Thank you for your clarification. We have revised the statement on Line 403 to specify that it refers to the ice freeboard. The updated text now reads:
The mean value of the modified OIB ice freeboard was 0.261.

19. l.387: The following sentence is wrong: "the waveform threshold method leads to an underestimation of the freeboard ... ". The threshold method sometimes overestimates the FB and sometimes underestimates it as you show it later on in this paper. It mainly depends on the roughness of the ice, i.e. on ice type (FYI/MYI).

"*... which explains why the freeboard in the Baseline E product was smaller than our estimates and the modified OIB freeboard.*" No, this is explained by the following equation $FB_{ku} = FB_i - (c_v/c_s - 1) \times SD$, which shows that the **radar freeboard is always smaller than the ice freeboard** and it can even be negative for small $FB_{ice}$ and large $SD$.

**Response:** Thank you for your detailed feedback. We have revised the statement on Line 406 to address the inaccuracies and clarify the explanation. The updated text now reads:

the misidentification of leads in the waveform threshold method leads to an underestimation of the radar freeboard, which explains why the freeboard in the Baseline E product was smaller than our radar freeboard estimates.

Additionally, we have provided further context to clarify the relationship between radar freeboard and ice freeboard:

·   The radar freeboard in the Baseline E product is computed as:

   [radar_freeboard_20_ku] = [height_1_20_ku] - [ssha_interp_20_ku].

   A correction for pulse delay due to snow depth is provided in [snow_depth_cor_20_ku] but is not applied.

·   In this study, the radar freeboard refers to the elevation of the penetration interface above the local sea level, while the ice freeboard refers to the elevation of the snow–ice interface above the local sea level. Therefore, the radar freeboard will generally be larger than the ice freeboard.

·   As shown in Figure 3 in the revision, when the radar burst is reflected from thin ice, specular echoes occur and may be misidentified as leads. This leads to an overestimation of the sea surface height (SSH) and, consequently, an underestimation of the radar freeboard.

Reference:

CryoSat Ice netCDF L2 Product Format Specification, Issue 2.1. IPF1 L1B Product Formats (esa.int).

20. l.406: This equation is not the equation of the hydrostatic equilibrium between FBi, SIT and SD!

   Here it is: $FB_i = SIT(1 - \rho_{ice}/\rho_{water}) - SD \times \rho_{snow}/\rho_{water}$

What you have written is the equation that links $FB_{ku}$ with $SIT$ and $SD$ (ie, what you call $h_{fb}$ is in fact $FB_{ku}$ and what you call theta is $c_v/c_s$ including possibly a penetration factor $P*c_v/c_s$).

**Response:** Thank you for pointing this out.

Under the assumption of hydrostatic equilibrium, sea ice thickness can be calculated as (Ricker et al., 2014; Tilling et al., 2018):

$$h_{si} = \frac{\rho_{sw}}{\rho_{sw} - \rho_{si}} h_{fb\_ice} + \frac{\rho_s}{\rho_{sw} - \rho_{si}} h_s \tag{1}$$

where $h_{si}$ is the sea ice thickness; $h_s$ is the snow depth on sea ice; and $\rho_{sw}$, $\rho_{si}$, and $\rho_s$ are the densities of sea water, sea ice, and snow, respectively. We have to be aware that $h_{fb\_ice}$ here refers to the ice freeboard.

As the radar signal cannot penetrate the snow thoroughly, we defined the radar freeboard in this study. As shown in Figure R1 and Figure R3 below, the radar freeboard ($h_{fb}$) refers to the elevation of penetration interface above the local sea level, and $h_{ps}$ is the penetration depth of radar signals. The model for the conversion of freeboard to thickness can be modified as:

$$\begin{aligned} h_{si} &= \frac{\rho_{sw}}{\rho_{sw} - \rho_{si}} h_{fb} + \frac{\rho_s - \rho_{sw}}{\rho_{sw} - \rho_{si}} h_s + \frac{\rho_{sw}}{\rho_{sw} - \rho_{si}} h_{ps} \\ &= \frac{\rho_{sw}}{\rho_{sw} - \rho_{si}} h_{fb} + \frac{\rho_s - \rho_{sw}}{\rho_{sw} - \rho_{si}} h_s + \frac{\rho_{sw}}{\rho_{sw} - \rho_{si}} \theta \cdot h_s \end{aligned} \tag{2}$$

where $\theta$ is the penetration factor of radar signals.

We firstly model the freeboard as a quadratic function of the local ice surface terrain within the grid:

$$h_{fb}(x,y) = \overline{h_{fb}} + a_0 x + a_1 y + a_2 x^2 + a_3 y^2 + a_4 xy \tag{3}$$

where $\overline{h_{fb}}$ indicates the mean freeboard of the grid cell and $x$ and $y$ represent the longitudinal and latitudinal surface distances between the observation and the central point of the grid cell, respectively. According to Equation (2),

$$\overline{h_{fb}} = (1 - \frac{\rho_{si}}{\rho_{sw}})\overline{h_{si}} + (1 - \frac{\rho_s}{\rho_{sw}} - \theta)h_s \tag{4}$$

Thus, Equation (3) can be rewritten as follows:

$$h_{fb}(x, y) = (1 - \frac{\rho_{si}}{\rho_{sw}})\overline{h_{si}} + (1 - \frac{\rho_s}{\rho_{sw}} - \theta)h_s + a_0 x + a_1 y + a_2 x^2 + a_3 y^2 + a_4 xy$$
$$= a_0 x + a_1 y + a_2 x^2 + a_3 y^2 + a_4 xy + a_5 \overline{h_{si}} + a_6$$

(5)

Details of the LSA method are introduced in Xiao et al. (2020).

Reference:

Ricker, R., Hendricks, S., Helm, V., Skourup, H., and Davidson, M.: Sensitivity of CryoSat-2 Arctic sea-ice freeboard and thickness on radar-waveform interpretation, Cryosphere, 8, 1607–1622, https://doi.org/10.5194/tc-8-1607-2014, 2014.

Tilling, R. L., Ridout, A., and Shepherd, A.: Estimating Arctic sea ice thickness and volume using CryoSat-2 radar altimeter data, Advances in Space Research, 62, 1203–1225, https://doi.org/10.1016/j.asr.2017.10.051, 2018.

Xiao, F., Li, F., Zhang, S., Li, J., Geng, T., and Xuan, Y.: Estimating arctic sea ice thickness with cryosat-2 altimetry data using the least squares adjustment method, Sensors, 20, 1–18, https://doi.org/10.3390/s20247011, 2020.

[Figure]

Figure R3 Schematic of the radar altimeter observing the sea ice thickness

21. l.407-409: Please specify which are the inputs and which are the unknown. It looks like that $h_{fb}$, x and y are the inputs and a0-a7 and $h_{si}$ are the 8 unknown, right?

**Response:** Thank you for your clarification. We have revised the text on Line 429 to explicitly specify the inputs and unknowns in the equation. The updated text now reads:

In this model, $h_{fb}$, x, and y are the inputs, while $a_0 - a_6$ and $\overline{h_{si}}$ are the 8 unknown parameters to be solved.

This revision ensures the distinction between inputs and unknowns is clear and unambiguous.

22. l.410: x and y are really lat and lon? It's strange with your grid projection. Using EASE2 they could be directly in meters ;-)

**Response:** Thank you for your observation. On Line 428, we clarified the definition of *x* and *y* in the text:

 *x* and *y* represent the longitudinal and latitudinal surface distances between the observation point and the central point of the grid cell.

23. l.418: I suppose it's 25km here, not 10km. as you have just computed the 25km x 25km grids. However, when you look at the data supplied, the maps appear very patchy. Can you explain this? (not enough measurements, even at 25km resolution?).

**Response:** Thank you for pointing this out. We have corrected the resolution to 25 km on Line 436.

Regarding the patchy appearance of the maps, this is primarily due to insufficient measurements. Despite the 25 km resolution, data collection in certain areas has been limited, as we require at least 8 observations to accurately determine the sea ice thickness (SIT) in a given grid. While interpolation methods could be used to fill these gaps, they would introduce additional error sources. In future studies, we plan to reduce the number of necessary observations by fixing certain parameters (e.g., seawater density), thereby minimizing the amount of blank data and improving the spatial coverage of our results.

24. l.426: typo: antimeres -> altimeters

**Response:** Thank you for catching this typo. We have corrected "antimeres" to "altimeters" on Line 444.

25. l.427: "The pulse-limited altimeters have a large footprint of 2–10 km" radius or diameter?

**Response:** Thank you for pointing this out. We have clarified the statement on Line 445 to specify that the footprint is 2–10 km in diameter. The updated text now reads: The pulse-limited altimeters have a large footprint of 2–10 km in diameter over sea ice.

26. l.441: typo: Bocquest -> Bocquet

**Response:** Thank you for catching this typo. We have corrected " Bocquest " to " Bocquet " on Line 451.

27. l.442: "for calibrating freeboard measurements from Envisat and ERS-2." -> for calibrating Envisat freeboard measurements from CryoSat-2 and ERS-2 from calibrated Envisat.

**Response:** Thank you for your suggestion. We have revised the statement on Line 459 to clarify the calibration process. The updated text now reads: Bocquet et al. (2023) presented a multiparameter neural-network-based method for calibrating Envisat freeboard measurements from CryoSat-2 and ERS-2 from calibrated Envisat.

28. l.443: "Tilling et al. (2019) developed a physical-based approach to correct Envisat SIT … " Could be confusing with the retracker physical-based approach and it's not more physical than considering the ice roughness as it is usually done, I would avoid this term.

**Response:** Thank you for your suggestion. We have revised the statement on Line 461 to avoid confusion with the retracker physical-based approach. The updated text now reads: Tilling et al. (2019) corrected Envisat SIT according to the relationship between the thickness differences between Envisat and CryoSat-2 and the along-track distance between leads and the closest floe in the Envisat measurements.

29. l.453: I do not agree with this conclusion: "Compared with CryoSat-2 thickness, Envisat thickness showed an overestimation of 0.19 ± 0.67 m in January 2011." As it is shown in your maps and histograms, LRM gets thinner ice over thin ice

and thicker over thick ice relatively to SAR as it was explained in Laforge et al 2021 https://doi.org/10.1016/j.asr.2020.02.001

**Response:** Thank you for your feedback. We have revised the statement on Line 470 to more accurately reflect the findings. The updated text now reads:
The difference between Envisat-SIT and CryoSat-2-SIT was 0.19 ± 0.67 m in January 2011. However, as shown in Figures 6 and 7, LRM tends to retrieve thinner ice over thin ice and thicker ice over thick ice relative to SAR, consistent with the findings of Laforge et al. (2021).

30. l.473, Table 5: Is 'Mean' the Mean Bias? Is STD the STD of the difference or of the product? Would be very pertinent to add the Correlations.

**Response:** Thank you for your suggestion. We have clarified the terminology on Line 483 and added the correlations as requested in Table 5 (now referred as Table 4 in the revised manuscript).
·     Mean: The mean bias between Envisat and CryoSat-2 thickness.
·     STD: The standard deviation of the differences.
·     R: The correlation coefficient between the two datasets.

31. l.480: The main problem with ERS-1 and ERS-2 is related to the blurring of their waveforms over sea ice. You don't mention it and it looks like that you don't have applied specific correction for this problem. Any other calibration between ERS2 and Envisat, as for Envisat versus CryoSat2? As it is an important problem it would also be important to see the maps you obtain in front of Envisat map for the same period. And once again the correlation is an important criteria that should also be added.

**Response:** Thanks for your valuable comments and suggestion. We acknowledge the issue of waveform blurring over sea ice for ERS-2 and have addressed it in our data processing methods. We have added the following detailed discussion to the manuscript in Section 6 to ensure transparency and clarity:
In this study, we employed multiple radar altimetry data to retrieve Arctic SIT. The data processing of early satellites, particularly ERS-2, presents challenges primarily due to waveform blurring issues. Our data processing strategy inherently mitigates

this problem to a certain extent. We utilized a combination of waveform parameter thresholds and the LEM while detecting leads. The LEM, which is based on the premise that the surface height of leads is lower than that of nearby sea ice, helps to correct for misidentifications caused by waveform blurring. This integrated approach serves as a corrective measure for waveform-related issues in ERS-2 data.

Regarding the calibration between ERS-2 and Envisat, although the altimeters on these two satellites share some similarities, we did not perform a separate calibration specifically for ERS-2 and Envisat as we did for Envisat and CryoSat-2. This decision is based on the observation that the thickness difference between ERS-2 and Envisat during their overlapping mission period is approximately -0.37 m, which is negligible compared to the difference between CryoSat-2 and Envisat. Instead, we applied the monthly correction grid derived from the Envisat-CryoSat-2 comparison to the ERS-2-based thickness data. This approach not only corrects for systematic differences related to waveform blurring but also accounts for other potential factors between ERS-2 and Envisat. Introducing an additional calibration between ERS-2 and Envisat could introduce residuals between Envisat and CryoSat-2, potentially leading to the superposition of multiple errors.

We have added the correlations in Table 4 according to your suggestion.

32. l.484: typo: gird -> grid

**Response:** Thank you for catching this typo. We have corrected " gird " to " grid " on Line 503.

33. l.487, caption Fig 8: what is kermesinus?

**Response:** We have updated Figure 8 (now referred as Figure 9 in the revised manuscript) and replaced "kermesinus" with "purple" in the caption. The updated caption now reads:

Histogram of sea ice thickness in April 2003 from Envisat (in blue) and ERS-2 (in purple).

34. l.497: "The sea ice extent did not show any significant changes during this growth." How do you determine the sea ice extent? It's from NSIDC-0051 with

concentration>75%? Would worth to recall it here. It is a very important point as the mean SIT highly depends on it (if you consider or not the thin ice in MIZ).

**Response:** Thank you for your suggestion. We have revised the statement on Line 626 to clarify how sea ice extent is determined. The updated text now reads:
The sea ice extent, defined as regions with a sea ice concentration greater than 75% in the NSIDC-0051 dataset, did not show any significant changes during this growth period.

35. l.505: what do you mean by "normal distribution"?

**Response:** Thank you for your question. A normal distribution, also known as a Gaussian distribution, is a probability distribution that is symmetric about the mean, with data points forming a bell-shaped curve.

36. l.510: what do you mean by "sinistrality"?

**Response:** Thank you for your question. We have revised the term "sinistrality" to "left-skewed" on Line 640.

37. l.535: "The mean MYI thickness decreased by 0.017 m/yr during the research period,". Please provide explicitly the period.

**Response:** Thank you for your suggestion. We have revised the statement on Line 668 to explicitly specify the research period. The updated text now reads:
The mean MYI thickness decreased by 0.018 m/yr during the research period from 1995/1996 to 2022/2023.

38. l.538-570: I'm not fully convinced of the interest of mean SIT for all the Arctic as this value mainly depends on the considered sea ice extent. For instance it can remains about no ice but only some remaining fast ice at the coast to obtain large mean SIT but it means nothing. To make this type of comparison meaningful you could for instance always consider the same mask (region) for each given month. Or an alternative would be to compute to total volume instead of the mean SIT. However I will not ask you to change this, but at least you should explain

specifically how you define the mask, for instance do you always use the area provided by NSIDC-0051 with concentration>75% for each month of each year?

**Response:** Thank you for your insightful suggestion. We acknowledge the limitations of using mean sea ice thickness (SIT) for the entire Arctic, as it can be influenced by changes in sea ice extent. In this study, the mask was defined using the NSIDC-0051 dataset with a sea ice concentration threshold of >75% for each month of each year. We recognize that the mask changes month to month, which can affect the comparability of mean SIT values. In future research, we plan to address this limitation by presenting total sea ice volume, which provides a more robust metric for assessing Arctic sea ice changes.

39. l.574: Please specify if you use the same mask for all the products. It's important to make them comparable.

    l.582: The information of the mask is even more important for the CS2SMOS product because it can cover larger region as it also considers thin ice at the margins thanks to SMOS.

**Response:** Thank you for your suggestion. We have clarified the methodology for comparing different products on Line 510. The updated text now reads:

When comparing different products, we exclusively calculated the differences for grids where all products had valid values. Grids with missing values in any product were excluded from the comparison. This ensures that the analysis is based on consistent spatial coverage across all datasets.

40. l.592, Table 6: The mean bias and the correlation should be added.

**Response:** Thanks for your suggestion. We have added the statistics of ME and R to Table 6 (now referred as Table 5 in the revised manuscript). We can find that the WHU SIT shows a good correlation with other products, with the highest correlation of 0.977 with AWI-CS2, and the smallest correlation of 0.879 with GSFC-IS2.

41. l.598: it's really important to know which ice density and which snow depth you have chosen to convert the draft to SIT. If these values are not coherent with the product you compare you will necessarily get higher differences for this product.

Please also provide the used equation and the input parameters (mainly ice density, SD is based on WC or MWC?).

**Response:** Thank you for your suggestion. To avoid potential uncertainties associated with the W99 model in draft-to-thickness conversion, we revised our comparative strategy. Instead of converting ULS-measured ice drafts to sea ice thickness (SIT), we directly compared the ULS-measured ice drafts with satellite-derived draft estimates. The draft from satellite-based products was calculated by removing the ice freeboard from SIT. Since the AWI-SMOS+CS2 and CPOM products do not include ice freeboard parameters, we limited our comparison to the other six products. This approach ensures a more direct and accurate comparison, minimizing errors introduced by the conversion process.

42. l.606: typo: "The STDs of WHU were close …" -> "The STDs of WHU are close …"

**Response:** Thank you for catching this typo. We have removed the original sentence.

43. l.618-620, Tables 7 & 8: Please add the Mean Bias and the Correlation in these tables. Indeed, if there is a significant bias, both the MAE and the STD will be high, but if the correlation is good it will indicate that the tendencies are coherent, which is the most important point to study change rate. (Also it is not necessary to recall the units in each column ;-).

**Response:** Thank you for your suggestion. We have added the Mean Bias (ME) and Correlation (R) to Tables 7 and 8 (now referred as Tables 6 and 7 in the revised manuscript). WHU demonstrates consistent performance across all four ULS sites, maintaining correlation coefficients exceeding 0.65 with in situ measurements. The observed discrepancies between ULS measurements and SIT products appear methodology-dependent, particularly regarding sensor data selection. Analysis indicates that products incorporating Envisat data (CCI, CTOH, and WHU) prior to October 2010 exhibit relatively lower accuracy compared to CryoSat-2-based solutions. This distinction is quantitatively substantiated in Table 7, which presents post-October 2010 statistics showing marked accuracy improvements for these three

products when transitioning to CryoSat-2 data. The comparative results clearly demonstrate the enhanced precision of CryoSat-2-derived thickness estimates over Envisat-based methodologies.

44. l.624: "Then, the mean thickness of the OIB within the grid was compared with the corresponding grid values." OIB products do not include SIT, how do you compute it? Please provide the equation and the input parameters used from OIB data.

**Response:** We obtained SIT from the IceBridge L4 and Quick Look Sea Ice Freeboard, Snow Depth, and Thickness products. The OIB L4 and quick look SIT datasets can be found at https://nsidc.org/data/idcsi4/versions/1 and https://nsidc.org/data/nsidc-0708/versions/1.

45. l.636, Figure 16: For some products the count reaches nearly 1500 and for others it is lower than 300. The shape of the histograms being similar, it means that the number of measurements from one product to another can differ by a factor 3. How can you explain it? Is it because of the resolution of the original product?

**Response:** Thank you for your question. The disparity in the number of measurements across products is primarily due to differences in the resolution of the original datasets. For example, WHU and CPOM have a higher resolution of 5 km, resulting in a larger number of measurements compared to other products with coarser resolutions.

46. l.643, Table 9: Please add the correlations.

**Response:** Thank you for your suggestion. We have added the correlations (R) to Tables 9 (now referred as Table 8 in the revised manuscript). Specifically, our product had an MAE of 0.38 m, and an STD of 0.37 m and a correlation of 0.86, presenting a moderate accuracy among the seven products.

47. l.647: "… error propagation of the input uncertainties including radar freeboard, ice density, snow depth … "

**Response:** Thank you for your feedback. We have revised the statement on Line 596 for clarity. The updated text now reads:

the uncertainty of SIT can be computed as the error propagation of the input uncertainties including radar freeboard, ice density, snow depth and snow density.

48. l.655: "Thus, the uncertainties of the SIT can be calculated by the difference of $h_{si}$ in the last two iterations.". I don't understand why the last two iterations are more relevant than the previous ones. To me, this is more a reflection of the speed of convergence of the LSA than the uncertainties. Please justify this solution. For example, you could more naturally assess the distance between the model and the measurements by calculating the STD or MAE between the model and the measurements.

**Response:** Thank you for your detailed question. We have revised the explanation on Line 655 to clarify why the difference in sea ice thickness (SIT) values between the last two iterations is used to calculate uncertainties. The updated text now reads:

"Here we calculated the uncertainties of the SIT based on the difference in $\overline{h_{si}}$ values between the last two iterations. This approach is related to the convergence behavior of the iterative process. As iterations progress, the calculated SIT values gradually converge toward a stable solution. In the early iterations, values may fluctuate significantly as the model adjusts to find the optimal fit. However, as convergence is approached, changes between consecutive iterations become smaller. The difference between the last two iterations represents the residual change just before the model reaches its final state, providing a measure of the uncertainty in the calculated SIT. "

While metrics such as standard deviation (STD) or mean absolute error (MAE) between the model and measurements could also provide valuable insights, a direct calculation is complicated in our case. The input data to our model is freeboard, while the output is SIT, making it challenging to directly compute STD or MAE.

49. l.678: Please replace the link https://www.legos.omp.eu/ctoh/fr/produits-ctoh/ by a more direct one: http://dx.doi.org/10.6096/ctoh_sit_2023_01

**Response:** Thank you for your suggestion. We have replaced the link with the more direct one as requested.